# Recognition Models to Learn Dynamics from Partial Observations with Neural ODEs

**Mona Buisson-Fenet**  *mona.buisson@minesparis.psl.eu*
*Ansys Research Team, Ansys France*
*Centre Automatique et Systèmes, Mines Paris - PSL University*
*Institute for Data Science in Mechanical Engineering - RWTH Aachen University*

**Valery Morgenthaler**  *valery.morgenthaler@ansys.com*
*Ansys Research Team, Ansys France*

**Sebastian Trimpe**  *trimpe@dsme.rwth-aachen.de*
*Institute for Data Science in Mechanical Engineering - RWTH Aachen University*

**Florent Di Meglio**  *florent.di_meglio@minesparis.psl.eu*
*Centre Automatique et Systèmes, Mines Paris - PSL University*

**Reviewed on OpenReview:** *https://openreview.net/forum?id=LTAdaRM29K*

## Abstract

Identifying dynamical systems from experimental data is a notably difficult task. Prior knowledge generally helps, but the extent of this knowledge varies with the application, and customized models are often needed. Neural ordinary differential equations can be written as a flexible framework for system identification and can incorporate a broad spectrum of physical insight, giving physical interpretability to the resulting latent space. In the case of partial observations, however, the data points cannot directly be mapped to the latent state of the ODE. Hence, we propose to design recognition models, in particular inspired by nonlinear observer theory, to link the partial observations to the latent state. We demonstrate the performance of the proposed approach on numerical simulations and on an experimental dataset from a robotic exoskeleton.

## 1 Introduction

Predicting the behavior of complex systems is of great importance in many fields. In engineering, for instance, designing controllers for robotic systems requires accurate predictions of their evolution. The dynamic behavior of such systems often follows a certain structure. Mathematically, this structure is captured by differential equations, e.g., the laws of physics. However, even an accurate model cannot account for all aspects of a physical phenomenon, and physical parameters can only be measured with uncertainty. Data-driven methods aim to enhance our predictive capabilities for complex systems based on experimental data.

We focus on dynamical systems and design an end-to-end method for learning them from experimental data. We investigate State-Space Models (SSMs), which are common in system theory, as many modern control synthesis methods build on them and the states are often amenable to physical interpretation. For many systems of interest, some degree of prior knowledge is available. It is desirable to include this knowledge into the SSM. To this end, we consider neural ordinary differential equations (NODEs), which were introduced by Chen et al. (2018) and have since sparked significant interest, e.g., Zhong et al. (2020); Rubanova et al. (2019). Their aim is to approximate a vector field that generates the observed data following a continuous-time ODE with a neural network. Their formulation is general enough to avoid needing a new design for each new system, but can also enforce a wide range of physical insight, allowing for a meaningful and interpretable

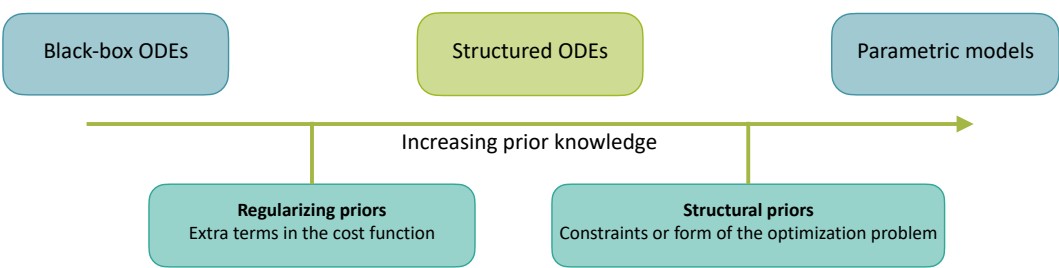

Figure 1: SSMs can include a broad spectrum of physical knowledge. On the left, purely data-based formulations such as latent NODEs are general but tend to violate physical principles and have trouble generalizing. On the right, parametric models can be identified from data: they extrapolate well but are system-specific and require expert knowledge. One can bridge this gap by including the available physical knowledge in an NODE formulation (2), in particular "regularizing" priors (extra terms in the cost function) or "structural" priors (constraints or form of the optimization problem).

model. Specific approaches have been proposed to include different priors, which we briefly recall; we present a unified view and include them in the proposed end-to-end framework.

Learning an SSM satisfying these priors amounts to learning the dynamics in a specific set of coordinates. However, experimental data is typically only partial, as not all of these coordinates, or *states*, are measured. This is a common problem in machine learning, where the existence of an underlying state that can explain the data is often assumed. In the systems and control community, estimating this underlying state from partial observations is known as state estimation or observer design. An observer is an algorithm that estimates the full latent state given a history of partial measurements and control inputs Bernard (2019); Bernard et al. (2022). While observer design provides the theoretical framework for latent state estimation, such as convergence guarantees of the estimate to the true state, it has not received much attention in the machine learning community. Hence, we propose to leverage concepts from nonlinear observer design to learn NODEs with physical priors from partial observations.

We design so-called *recognition models* to map the partial observations to the latent state. We discuss several approaches, in particular based on a type of nonlinear observers called Kazantzis-Kravaris/Luenberger (KKL) observers (Kazantzis & Kravaris, 1998; Andrieu & Praly, 2006). We show that the KKL-based recognition models perform well and have desirable properties, e.g., a given size for the internal state. Such recognition models can then be embedded in the NODE formulation or any other optimization-based system identification algorithm. Our main contributions can be summarized as follows:

- We formulate structured NODEs as a flexible framework for learning dynamics from partial observations, which enables enforcing a broad spectrum of physical knowledge;
- We introduce recognition models to link observations and latent state, then propose several forms based on nonlinear observer design;
- We compare the proposed recognition models in a simulation benchmark;
- We apply the proposed approach to an experimental dataset obtained on a robotic exoskeleton, illustrating the possibility of learning a physically sound model of complex dynamics from real-world data.

Combining these yields an end-to-end framework for learning physical systems from partial observations.

## 2  Related work

The proposed method is based on two research areas: nonlinear observer design and machine learning for dynamical systems. We give an overview of the main trends and the most related methods on these topics.

### 2.1  System theory

In system theory, many subfields are concerned with the study of dynamical systems from experimental data.

**System identification**   The area of system identification aims at finding a possible dynamics model given a finite amount of partial measurements (Ljung, 1987; Nelles, 2001; Schoukens & Ljung, 2019). For linear systems, a suitable set of system matrices can be identified using subspace methods (Viberg, 1995). For nonlinear systems, most state-of-the-art techniques aim at estimating the variables of a given parametric model using Bayesian parameter estimation (Galioto & Gorodetsky, 2020) or optimization-based methods (Schittkowski, 2002; Raue et al., 2013; Villaverde et al., 2021), or a decomposition of its dynamics on a suitable basis of functions (Sjöberg et al., 1995). These classical methods tend to be system-specific: they require expert knowledge to construct a parametric model or precisely pick the hypothesis class in which it will be approximated. NODEs are a general tool for system identification in case no parametric model is available, in which a broad range of physical knowledge can be included by adapting the formulation.

**Observer design**   When identifying a state-space model from partial observations, the unknown latent state must be estimated. This is the objective of state observers or estimators, which infer the state from observations by designing an auxiliary system driven by the measurement (see Bernard (2019); Bernard et al. (2022) for an overview). Observers often assume an accurate dynamics model, but designs that can deal with imperfect models are also available. In that case, the unknown parts of the dynamics can be overridden through high-gain or sliding-mode designs to enable convergence (Buisson-Fenet et al., 2021; Shtessel et al., 2016). Otherwise, the unknown parameters can be seen as extra states with constant dynamics, and extended state observers can be designed, such that the estimated state and parameters converge asymptotically (Praly et al., 2006). Some concepts from observer theory can be leveraged to improve upon existing approaches for learning dynamics from partial observations, which require estimating the unknown latent state.

## 2.2   Learning dynamical systems

Learning dynamical systems from data is also investigated in machine learning (Legaard et al., 2021; Nguyen-Tuong & Peters, 2011). We focus on settings considered realistic in system identification, i.e., methods that allow for control and partial observations, and can ensure certain physical properties.

**Physics-aware models**   The dynamics models obtained from machine learning often struggle with generalization and do not verify important physical principles such as conservation laws. Therefore, there have been efforts to bring together machine learning and first principles to learn physics-aware models; see Wang & Yu (2021) for an overview of these efforts in deep learning. In general, there are two takes on including physical knowledge in data-driven models, as illustrated in Fig. 1. On the one hand, "regularizing" priors can be included by adding terms to the cost function to penalize certain aspects. The most common case is when a prior model of the system is available from first principles. We can then learn the residuals of this prior model, i.e., the difference between the prior predictions and the observations, while penalizing the norm of the residual model to correct the prior only as much as necessary. This is investigated in Yin et al. (2021); Mehta et al. (2020) for full state observations. Other quantities can be known a priori and enforced similarly, such as the total energy of the system (Eichelsdörfer et al., 2021) or stability through a Lyapunov-inspired cost function (Schlaginhaufen et al., 2021). On the other hand, structural properties can be enforced by constraints or a specific form of the optimization problem. This yields a harder problem, but improves the performance and interpretability of the model. This line of work originates from Lutter et al. (2019); Greydanus et al. (2019); Cranmer et al. (2020) (see Zhong et al. (2021) for an overview) and has been extended to NODEs for Hamiltonian and port-Hamiltonian systems (Zhong et al., 2020; Massaroli et al., 2020a; Zakwan et al., 2022), but also to enforce more general energy-based structures (Manek & Zico Kolter, 2019; Course et al., 2020) or the rules of electromagnetism (Zhu et al., 2019). However, little previous work on NODEs assumes partial and noisy measurements of system trajectories.

There exist various other methods to learn physics-aware dynamics models, such as Bayesian approaches, in which prior knowledge can be enforced in the form of the kernel (Wu et al., 2019), by structural constraints (Geist & Trimpe, 2021; Rath et al., 2021; Ensinger et al., 2022), or by estimating the variables of a parametric model (Galioto & Gorodetsky, 2020). In this paper, we focus on NODEs, which leverage the predictive power of deep learning while enforcing a broad range of physical knowledge in the problem formulation.

**Partial observations** Most NODE frameworks for dynamical systems assume full state measurements. Partial observations greatly increase the complexity of the problem: the latent state is unknown, leading to a large number of possible state-space representations. In this case, the question of linking the observations with the latent state needs to be tackled. In Bayesian approaches, the distribution over the initial state can be directly conditioned on the first observations then approximated by a so-called recognition model (Eleftheriadis et al., 2017; Doerr et al., 2017; 2018). Such an approach has also been used for Bayesian extensions of NODEs, where the NODE describes the dynamics of the latent state while the distribution of the initial latent variable given the observations and vice versa are approximated by encoder and decoder networks (Yildiz et al., 2019; Norcliffe et al., 2021a). The encoder network, which links observations to latent state by a deterministic mapping or by approximating the conditional distribution, can also be a Recurrent Neural Network (RNN) (Rubanova et al., 2019; Doyeon et al., 2021; de Brouwer et al., 2019; Rubanova et al., 2019) or an autoencoder (Bakarji et al., 2022). The particular case in which the latent ODE is linear and evolves according to the Koopman operator (that can be jointly approximated) is investigated in Lusch et al. (2018); Bevanda et al. (2021). In general, little insight into the desired latent representation is provided. This leads to difficulties for the obtained models to generalize, but also with their interpretability: often in a control environment, the states should have a physical meaning. Therefore, we propose to learn a recognition model that maps the observations to the latent state, while enforcing physical knowledge in the latent space.

## 3 Problem statement

Consider a general continuous-time nonlinear system

$$\dot{x}(t) = f(x(t), u(t)) \qquad\qquad y(t) = h(x(t), u(t)) + \epsilon(t) \qquad\qquad (1)$$
$$x(0) = x_0$$

where $x(t) \in \mathbb{R}^{d_x}$ is the state, $u(t) \in \mathbb{R}^{d_u}$ is the control input, $y(t) \in \mathbb{R}^{d_y}$ is the measured output, and $f$, $h$ are the true dynamics and measurement functions, assumed continuously differentiable. We denote $\dot{x}(t)$ the derivative of $x$ w.r.t. time $t$, and generally omit the time dependency. We only have access to partial measurements $y$ corrupted by noise $\epsilon$, the control input $u$, and the measurement function $h$: the dynamics $f$ and the state $x$ are unknown. We assume that the solutions to (1) are well-defined and aim to estimate $f$.

The aim of NODEs (Chen et al., 2018) is to learn a vector field that generates the data through an ODE, possibly up to an input and output transformation; see Massaroli et al. (2020b) for an overview. While the interpretation of this formulation for general machine learning tasks remains open, it is very natural for SSMs: it amounts to approximating the dynamics with a neural network. However, there is no unifying framework for applying NODEs to dynamical systems in realistic settings, i.e., with partial and noisy trajectory data, with a control input, and using all available physical knowledge; we present one in this paper.

Assume we have access to $N$ measured trajectories indexed by $j$, denoted $\underline{y}^j$ and sampled at times $t_i$, $i \in \{0, ..., n-1\}$. We approximate the true dynamics $f$ with a neural network $f_\theta$ of weights $\theta$. If the initial conditions $x_0^j$ are known, learning $f_\theta$ can be formulated as the following optimization problem:

$$\min_\theta \quad \frac{1}{2d_y n N} \sum_{j=1}^{N} \sum_{i=0}^{n-1} \left\| y^j(t_i) - \underline{y}^j(t_i) \right\|_2^2 \qquad\qquad (2)$$
$$s.t. \quad \dot{x}^j = f_\theta(x^j, u^j) \qquad y^j = h(x^j, u^j)$$
$$x^j(0) = x_0^j,$$

where the constraint is valid for all $j \in \{1, ..., N\}$. Several methods have been proposed to compute the gradient of (2); see Schittkowski (2002); Alexe & Sandu (2009); Massaroli et al. (2020b) for details. We opt for automatic differentiation through the numerical solver (`torchdiffeq` by Chen et al. (2018)).

This problem is not well-posed: for a given state trajectory, there exist several state-space representations $f$ that can generate the data. This is known as the unidentifiability problem (Aliee et al., 2021). The key problems to obtain meaningful solutions to (2) are (i) enforcing physical knowledge to learn a state-space

representation that not only explains the data, but is also physically meaningful, and (ii) dealing with partial observations, i.e., unknown latent state and in particular unknown $x_0$. Problem (i) has been addressed in the literature for some particular cases, as presented in Sec. 2.2, by including the available physical knowledge in the form of "regularizing" or "hard" priors (see Fig. 1). We denote the general approach of adapting (2) to build physics-aware NODEs as *structured NODEs*, and apply it to examples with varying prior knowledge.

Problem (ii) remains largely open. For fixed $f_\theta$ and $u$, each prediction made during training for a given measured trajectory is determined by the corresponding $x_0$; hence, estimating this initial state is critical. We tackle (ii) by designing recognition models in Sec. 4, which is our main technical contribution. We combine them with structured NODEs to simultaneously address (i) and (ii). This yields an end-to-end framework for system identification based on physical knowledge and partial, noisy observations, a common setting in practice; we apply it to several examples in Sec. 5. While some of the components needed for this framework exist in the literature, e.g., on building physics-aware NODEs (Sec. 2.2), the vision of recognition models, the presentation in a unified framework and the application to relevant practical cases are novel.

**Remark 1** *The considered setting is generally regarded as realistic in system identification, since $y$ is measured by sensors, and $u$ is chosen by the user, who often knows which part of the state is being measured. However, if that is not the case, it is always possible to train an output network $h_\theta$ jointly with $f_\theta$.*

## 4 Recognition models

In the case of partial observations, the initial condition $x_0$ in (2) is unknown and needs to be estimated jointly with $f_\theta$. Estimating $x_0$ from partial observations is directly related to state estimation: while observers run forward in time to estimate the state asymptotically, we formulate this recognition problem as running backward in time to estimate the initial condition. Therefore, the lens of observer design is well suited for investigating recognition models, though it has not been often considered. For example, whether the state can be estimated from the output is a precise notion in system theory called observability (Bernard, 2019):

**Definition 1** *Initial conditions $x_a$, $x_b$ are uniformly distinguishable in time $t_c$ if for any input $u : \mathbb{R} \mapsto \mathbb{R}^{d_u}$*

$$y_{a,u}(t) = y_{b,u}(t) \ \forall \ t \in [0, t_c] \Rightarrow x_a = x_b, \tag{3}$$

*where $y_{a,u}$ (resp. $y_{b,u}$) is the output of (1) given input $u$ and initial condition $x_a$ (resp. $x_b$). System (1) is observable (in $t_c$) if all initial conditions are uniformly distinguishable.*

Hence, if (2) is observable, then $x(0)$ *is uniquely determined by $y$ and $u$ over $[0, t_c]$ for $t_c$ large enough.* This assumption is necessary, otherwise there is no hope of learning $f_\theta$ from the observations only.

In system identification, the unknown initial condition is usually optimized directly as a free variable: it needs to be optimized again for each new trajectory and cannot be used as such for prediction. Instead, we propose to estimate it from the observations by learning a recognition model $\psi_\theta$ that links the output to the initial state. We design $\psi_\theta$ as a neural network and denote its input $\bar{z}(t_c)$, to be described in the following. Concatenating the weights of $f_\theta$ and $\psi_\theta$ into $\theta$ leads to the modified problem:

$$\min_\theta \quad \frac{1}{2d_y n N} \sum_{j=1}^{N} \sum_{i=0}^{n-1} \left\| y^j(t_i) - \underline{y}^j(t_i) \right\|_2^2 \tag{4}$$

$$s.t. \quad \dot{x}^j = f_\theta(x^j, u^j) \qquad y^j = h(x^j, u^j)$$
$$x^j(0) = \psi_\theta(\bar{z}^j(t_c)).$$

### 4.1 General approaches

Some recognition methods have been proposed in the literature, not necessarily for system identification with NODEs, rather for probabilistic (Doerr et al., 2017) or generative Yildiz et al. (2019) models. We draw inspiration from them and rewrite them to fit into our general framework, leading to the following.

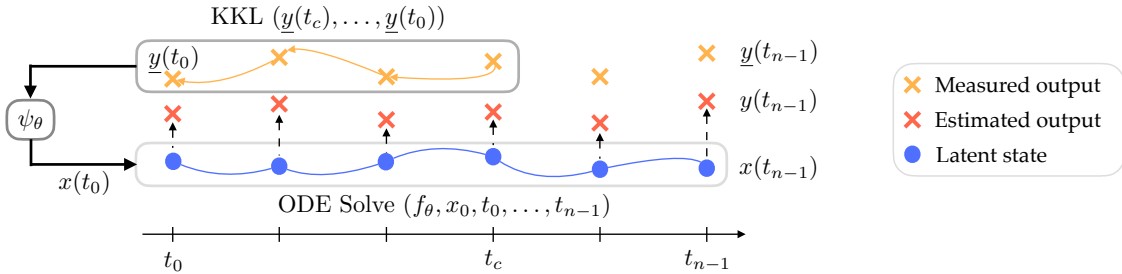

Figure 2: Illustration of the method: the KKL observer runs backward over the observations on $[t_c, 0]$, the initial latent state is estimated, then the NODE runs forward and predicts the following trajectory.

**Direct method**    The most straightforward approach is to stack the observations and learn a mapping from

$$\bar{z}(t_c) = \underline{y}_{0:t_c} := (\underline{y}(0), \dots, \underline{y}(t_c)) \tag{5}$$

to the initial latent state. For nonautonomous systems, the first inputs should also be taken into account, which yields $\bar{z}(t_c) = (\underline{y}_{0:t_c}, u_{0:t_c})$. We denote this as the direct method. Variants of this approach has been used for approximating the distribution over the initial state conditioned on $\underline{y}_{0:t_c}$, e.g., for joint inference and learning of Gaussian process state-space models (Eleftheriadis et al., 2017; Doerr et al., 2017; 2018). Augmentation strategies for NODEs (Dupont et al., 2019; Massaroli et al., 2020b; Chalvidal et al., 2021; Norcliffe et al., 2021b) are often particular cases with $t_c = 0$. However, for many nonlinear systems, this is too little information to estimate $x(0)$. There are few works on NODE-based system identification from partial observations, some of which train a recognition model from $\underline{y}_{-t_c:0}$ (Ayed et al., 2020; Yildiz et al., 2019; Norcliffe et al., 2021a), or learn the dynamics of $\underline{y}_{0:t_c}$ (Schlaginhaufen et al., 2021).

As justified by the observability assumption, for $t_c$ large enough this is all the information needed to estimate $x(0)$. However, for large $t_c$, the input dimension becomes arbitrarily high, so that optimizing $\psi_\theta$ is more difficult. Also, the observations are not preprocessed in any way, though they may be noisy.

**Recurrent recognition models**    Latent NODEs (Chen et al., 2018; Rubanova et al., 2019) also use a recognition model to estimate the initial latent state from observations, though this may not be the state of an SSM. This recognition model is a Recurrent Neural Network (RNN) in Chen et al. (2018) and a RNN combined with a second NODE model in Rubanova et al. (2019). These methods filter the information contained in $(\underline{y}_{0:t_c}, u_{0:t_c})$ in backward time then feed it to the recognition network $\psi_\theta$, which is trained jointly with the (ODE-)RNN. We consider this baseline in the numerical results: we combine a Gated Recurrent Unit (GRU) of internal dimension $d_z$, run in backward time so that $\bar{z}(t_c)$ is the output of the GRU, and an output network $\psi_\theta$. We denote this method from Chen et al. (2018) as RNN+.

We now propose a novel type of recognition model based on nonlinear observer design, leading to different choices of $\bar{z}(t_c)$. See Table 1 for a summary of the proposed recognition methods.

## 4.2   KKL-based recognition models

Nonlinear observer design is concerned with the estimation of the state of nonlinear SSMs from partial observations; see e.g., Bernard et al. (2022); Bernard (2019) for an overview. A particular method that has recently gained interest is the Kazantzis-Kravaris/Luenberger (KKL) observer (Kazantzis & Kravaris, 1998; Andrieu & Praly, 2006). Intuitively, KKL observers rely on building a linear filter of the measurement: an auxiliary dynamical system of internal state $z$ with known dimension $d_z$ is simulated, taking the measurement as input and filtering it to extract the information it contains. The observer state verifies

$$\dot{z} = Dz + F\underline{y} \qquad z(0) = z_0 \tag{6}$$

where $z \in \mathbb{R}^{d_z}$ with $d_z = d_y(d_x + 1)$ and $z_0$ is an arbitrary initial condition. In this system, $\underline{y}$ is the continuous-time measurement from (1), or alternatively an interpolation between the discrete observations $\underline{y}(t_i)$. The

parameters $D$ and $F$ are chosen such that $D$ is Hurwitz, i.e., all eigenvalues are in the left half-plane, and $(D, F)$ is a controllable pair, i.e., the matrix $\begin{pmatrix} F & DF & \dots & D^{d_z-1}F \end{pmatrix}$ has full rank (Kailath, 1980). Thanks to the stability of (6), the internal state $z$ "forgets" its arbitrary initial condition $z_0$ and converges asymptotically to a value that is uniquely determined by the history of $y$. Under certain conditions, this value uniquely determines, in turn, the value of the unmeasured state $x$.

More precisely, if there exists an injective transformation from $x$ to $z$, denoted $\mathcal{T}$, and its left inverse $\mathcal{T}^*$, then for any arbitrary $z_0$, the estimate $\hat{x}(t) = \mathcal{T}^*(z(t))$ converges to $x(t)$. This yields that $x(t) \simeq \mathcal{T}^*(x(t))$ for $t$ large enough. The existence of $\mathcal{T}$ is studied separately for autonomous and nonautonomous systems, under mild assumptions, mainly $x(t) \in \mathcal{X}$ compact $\forall\, t$ and (1) is backward distinguishable, i.e., Definition 1 in backward time[1]: the current state $x(t)$ is uniquely determined by $y$ and $u$ over $[t - t_c, t]$ for some $t_c$.

**Autonomous systems**  For autonomous systems, i.e., $u = 0$, it is shown by Andrieu & Praly (2006) that if the eigenvalues of $D$ have sufficiently large negative real parts, then there exists an injective transformation $\mathcal{T}$ and its left inverse $\mathcal{T}^*$ such that[2]

$$\|x(t) - \mathcal{T}^*(z(t))\| \xrightarrow[t \to \infty]{} 0, \tag{7}$$

meaning that $\mathcal{T}^*(z(t))$ with $z(t)$ from (6) is an observer for $x(t)$. However, $\mathcal{T}^*$ cannot be computed analytically in general. Therefore, it has been proposed to learn it from full-state simulations (da Costa Ramos et al., 2020; Buisson-Fenet et al., 2022) or to directly learn an output predictor $h \circ \mathcal{T}^*$ from partial observations (Janny et al., 2021). Running the observer (6) backward in time[3] on $\underline{y}_{t_c:0}$ yields $x(0) \approx \mathcal{T}(z(0)))$ for $t_c$ large enough. Hence, we propose to train a recognition model $x(0) = \psi_\theta(z(0))$, where $z(0)$ is the result of (6) run backward in time for $t_c$ from an arbitrary initial condition $z(t_c)$. This is further denoted as the KKL method, illustrated in Fig. 2 (see Table 1 for a summary of the proposed methods).

**Nonautonomous systems**  When extending the previous results to nonautonomous systems, $\mathcal{T}$ not only depends on $z(t)$ but also on time, in particular on the past values of $u$, and becomes injective for $t \geq t_c$ with $t_c$ from the backward distinguishability assumption (Bernard & Andrieu, 2019). In the context of recognition models, this dependency on $u$ over $[0, t]$ can be made explicit by running the observer (6) in backward time then training a recognition model $x(0) = \psi_\theta(\bar{z}(t_c))$ with $\bar{z}(t_c) = (z(0), u_{t_c:0})$. This is still denoted as the KKL method, for nonautonomous systems.

If the signal $u$ can be represented as the output of an auxiliary system of inner state $\omega$ with dimension $d_\omega$, it is shown in Spirito et al. (2022) that the static observer

$$\dot{z} = Dz + F \begin{pmatrix} y \\ u \end{pmatrix} \qquad\qquad z(0) = z_0 \tag{8}$$

leads to the same results as for autonomous systems. The time dependency in $\mathcal{T}$ disappears at the cost of a higher dimension: $d_z = (d_y + d_u)(d_x + d_\omega + 1)$. This functional approach leads to an alternative recognition model denoted KKLu: $x(0) = \psi_\theta(z(0))$ where $z$ is solution of (8) simulated backward in time, and $d_\omega$ is chosen large enough to generate $u$ (e.g., $d_\omega = 3$ for a sinusoidal $u$).

**Optimizing D jointly**  With any KKL-based recognition model, the choice of $D$ in (6) – resp. (8) – is critical since it controls the convergence rate of $z$. Hence, we propose to optimize $D$ jointly with $\theta$, as in Janny et al. (2021). More details are provided in the supplementary material.

For $t_c$ large enough, the transformation $\mathcal{T}^*$ approximated by $\psi_\theta$ is guaranteed to exist for a known dimension $d_z$. The KKL observer also filters the information and provides a low-dimensional input to $\psi_\theta$, which is expected to be easier to train. The RNN-based recognition models are close to learning a discrete-time observer with unknown dynamics: this is similar, but provides no theoretical argument for choosing the internal dimension of the RNN, no guarantee for the existence of a recognition model in form of an RNN, no physical interpretation for the behavior of the obtained observer, and leads to many more free parameters.

---

[1] If the solutions of (1) are unique, e.g., if $f$ is $C^1$, then distinguishability and backward distinguishability are equivalent.
[2] See the Appendix for technical details.
[3] We simulate $z$ backward in time, so that all samples can be used for data fitting after being inputted to $\psi_\theta$.

| Method | $\bar{z}(t_c)$ for autonomous | $\bar{z}(t_c)$ for nonautonomous |
|---|---|---|
| $t_c = 0$ | $\underline{y}(0)$ | $(\underline{y}(0), u(0))$ |
| direct | $\underline{y}_{0:t_c}$ | $(\underline{y}_{0:t_c}, u_{0:t_c})$ |
| RNN+ | GRU over $\underline{y}_{t_c:0}$ | GRU over $(\underline{y}_{t_c:0}, u_{t_c:0})$ |
| KKL | KKL over $\underline{y}_{t_c:0}$ | KKL over $\underline{y}_{t_c:0}$, concatenated with $u_{0:t_c}$ |
| KKLu | n/a | functional KKL over $(\underline{y}_{t_c:0}, u_{t_c:0})$ |

Table 1: Summary of the proposed recognition methods for autonomous and nonautonomous systems, all run backward in time. The recognition model $\psi_\theta$ is trained with $x(0) = \psi_\theta(\bar{z}(t_c))$; see Sec. 4 for details.

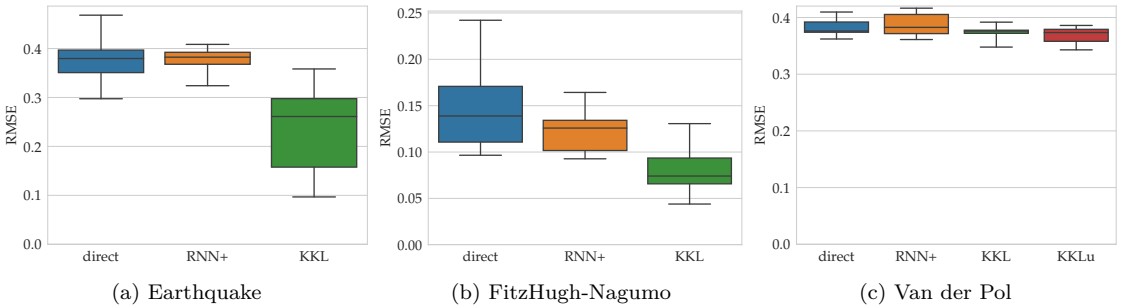

(a) Earthquake  (b) FitzHugh-Nagumo  (c) Van der Pol

Figure 3: Learning NODEs without structure, with recognition models direct, RNN+, KKL, and KKLu for the last, nonautonomous system. We compare the RMSE over the predicted output for 100 test trajectories.

## 5 Experiments

We demonstrate the ability of the proposed approach to learn dynamics from partial observations with varying degrees of prior knowledge, illustrated in Fig. 1. We first compare the different recognition models in combination with NODEs without priors. We then provide an extensive case study on the harmonic oscillator, often used in the literature, learning its dynamics from partial measurements with increasing priors. Finally, we apply our approach to a real-world, complex use case obtained on a robotic exoskeleton[4]. All models are evaluated w.r.t. their prediction capabilities: given $(y_{0:t_c}, u_{0:t_c})$ for a number of test trajectories, we estimate the initial state, predict the further output, then compute the RMSE over this predicted output.

### 5.1 Benchmark of recognition models

We demonstrate that the proposed recognition models can estimate the initial condition of a system from partial and noisy observations. We simulate three systems, the underlying physical state serves as ground truth. The first is a simplified model of the dynamics of a two-story building during an earthquake (Winkel, 2017; Karlsson & Svanström, 2019) with $d_x = 4$, $d_y = 1$. The second is the FitzHugh-Nagumo model, a simplified representation of a spiking neuron subject to a constant stimulus (Clairon & Samson, 2020) with $d_x = 2$, $d_y = 1$. The third is the Van der Pol oscillator with $d_x = 2$, $d_y = 1$. More details are provided in the supplementary material. The dynamics models are free: this corresponds to the left end of Fig. 1.

We train ten direct, RNN+, KKL and KKLu recognition models as presented in Sec. 4.2 and in Table 1. The recognition models estimate $x(0)$ from the information contained in $\underline{y}_{0:t_c}$ for the first, $(\underline{y}_{0:t_c}, u_0)$ for the second system, where $u_0$ is the value of the stimulus, and $(\underline{y}_{0:t_c}, u_{0:t_c})$ for the third system. We use $N = 50$ trajectories of 3 seconds; the output is corrupted by Gaussian noise, and the hyperparameters are chosen to be coherent between the methods and enable a fair comparison. For evaluation, we randomly select 100 test trajectories (also 3 s) and compute the RMSE over the predicted output. The results presented in Fig. 3 indicate that the more compressed structure from observer design helps build a more effective recognition

---

[4]Implementation details are provided in the supplementary material, code to reproduce the experiments is available at https://anonymous.4open.science/r/structured_NODEs-7C23.

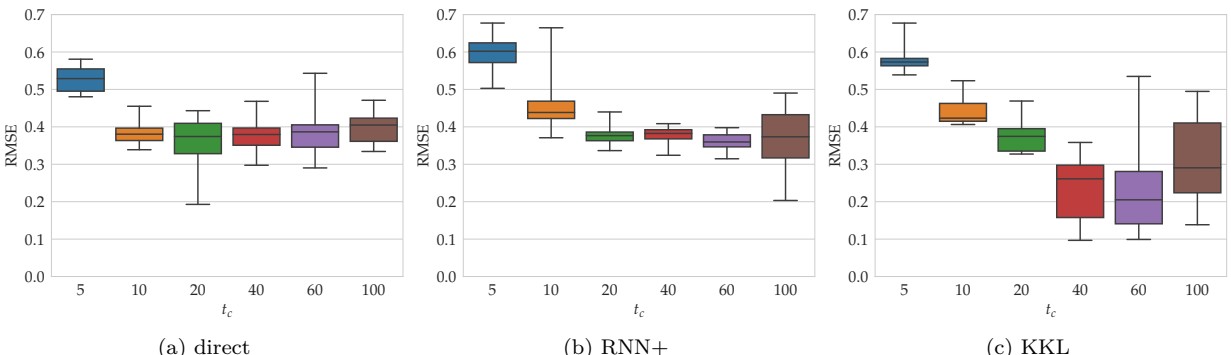

(a) direct        (b) RNN+        (c) KKL

Figure 4: NODE and recognition model for the earthquake model, for different lengths of $t_c$ (in time steps).

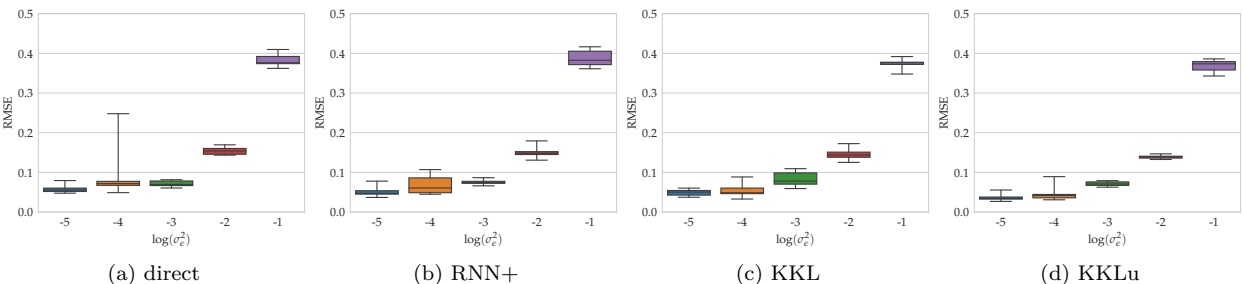

(a) direct     (b) RNN+     (c) KKL     (d) KKLu

Figure 5: NODE and recognition model for the Van der Pol oscillator, for different values of $\sigma_\epsilon^2$.

model. The direct method with $t_c = 0$ was run but is not shown here since it leads to much higher error, as it does not verify the observability assumption.

### 5.1.1 Ablation studies

In the previous benchmark, we choose all hyperparameters such that the comparison is as fair as possible. For example, $\psi_\theta$ is the same neural network, the dimension of the internal recognition state is the same for the RNN+ and KKL baselines ($d_z$ of the standard KKL for autonomous systems, $d_z$ of the functional KKL for nonautonomous systems). We now investigate the impact of two of the main parameters on the performance of each approach for the full NODE models: $t_c$ and the variance $\sigma_\epsilon^2$ of the Gaussian measurement noise.

For the study on $t_c$, we focus on the earthquake system. We run the same experiments as before with $t_c$ in $\{5, 10, 20, 40, 60, 100\} \times \Delta t$, where $\Delta t = 0.03$ s. As depicted in Fig. 4, when $t_c$ is too low it becomes difficult to estimate $x(0)$ from the information contained in $\underline{y}_{0:t_c}$: the system is not necessarily observable. It seems the threshold of observability is around $t_c = 30\Delta t = 0.9$ s, since the RMSE over the test trajectories stabilizes for higher values. For this system, the KKL method reaches the lowest error and keeps improving for higher values of $t_c$: the higher $t_c$, the more the observer has converged, the closer the relationship $x(0) \approx \mathcal{T}^*(\bar{z}(0))$ is and the easier it seems to learn $\psi_\theta$. For the other methods, $t_c$ seems to have less influence once the threshold of observability is reached since there is no notion of convergence over time.

For the study on $\sigma_\epsilon^2$, we focus on the Van der Pol oscillator. We test for values in $\{10^{-5}, 10^{-4}, 10^{-3}, 10^{-2}, 10^{-1}\}$ and obtain the results in Fig. 5. As expected, the higher the measurement noise variance, the higher the prediction error on the test trajectories. We again observe a threshold effect, under which further reduction of the noise variance leads to little improvement in the prediction accuracy. Note that for the KKL-based methods, we optimized $D$ once for each noise level from the same initial value, then used this optimized value for all ten experiments. If $D$ is only optimized for a specific noise level, then the performance is degraded at the others, for which this value might filter too much or too little.

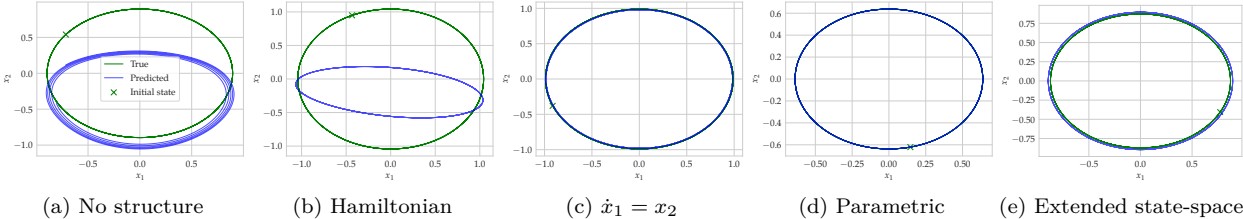

| (a) No structure | (b) Hamiltonian | (c) $\dot{x}_1 = x_2$ | (d) Parametric | (e) Extended state-space |

Figure 6: Structured NODEs for the harmonic oscillator, with KKL recognition and increasing priors. The true trajectory is in green, the prediction of a long trajectory (30 s) in blue to illustrate the long-term accuracy. Increasing structure yields a more interpretable, but also more accurate model, except for (e) which solves a more open problem (a new frequency is estimated for each trajectory).

| Method | (a) | (b) | (c) | (d) | (e) |
|---|---|---|---|---|---|
| direct | 0.040 (0.011) | 0.050 (0.033) | **0.035** (0.008) | **0.029** (0.005) | 0.080 (0.041) |
| RNN+ | 0.057 (0.014) | 0.055 (0.012) | 0.048 (0.003) | 0.037 (0.006) | 0.052 (0.011) |
| KKL | **0.036** (0.010) | **0.042** (0.011) | **0.036** (0.004) | 0.032 (0.003) | **0.049** (0.003) |

Table 2: Recognition models for the harmonic oscillator. We train ten models for each setting, then compute the median and interquartile range (in parentheses) of the RMSE on the predicted output for hundred test trajectories of 9 s. In most cases, KKL recognition leads to more accurate predictions.

## 5.2 Harmonic oscillator with increasing priors

We now illustrate how NODEs with KKL-based recognition can be combined with physics-aware approaches to cover different degrees of structure, as illustrated in Fig. 1. We simulate an autonomous harmonic oscillator with unknown frequency:

$$\dot{x}_1 = x_2 \qquad \dot{x}_2 = -\omega^2 x_1, \tag{9}$$

where $\omega^2 > 0$ is the unknown frequency of the oscillator and $y = x_1$ is measured, corrupted by Gaussian noise of standard deviation $\sigma = 0.01$. Various designs have been proposed to identify both the state and the model, i.e., the frequency, for example subspace methods, or an extended state-space model with a nonlinear observer; see Praly et al. (2006) and references therein. We demonstrate that our unifying framework can solve this problem while enforcing increasing physical knowledge. The results are illustrated in Table. 2 with a KKL recognition model, $\omega = 1$, $N = 20$ trajectories of 3 s for training and 100 trajectories of 9 s for testing.

First, the NODE is trained without any structure (a) as in (4), which leads to one of many possible state-space models: it fits the observations in $x_1$ but finds another coordinate system for the unmeasured state, as expected for general latent NODEs. It also does not conserve energy. Then, we enforce a Hamiltonian structure (b) by directly learning the Hamiltonian function $H_\theta(x)$. This leads to a dynamics model again in another coordinate system, but that conserves energy: we learn the dynamics up to a symplectomorphism (Bertalan et al., 2019). We then impose $\dot{x}_1 = x_2$ and only learn $\dot{x}_2 = -\nabla H_\theta(x_1)$ (c). This enforces a particular choice of Hamiltonian dynamics, such that the obtained model conserves energy and stays in the physical coordinate system ($x_1$ position, $x_2$ velocity). Imposing even more structure, we only optimize the unknown frequency $\omega^2$ jointly with the recognition model, while the rest of the dynamics are considered known (d). Another possibility is to consider the extended state-space model where $x_3 = \omega^2$ has constant dynamics (e). In that case, only a recognition model needs to be trained; however, $x(0) \in \mathbb{R}^3$ including the frequency is estimated for each trajectory, such that this formulation is much more open. Both (d) and (e), which correspond to the right end of the spectrum in Fig. 1, also lead to energy-conserving trajectories in the physical coordinates. The results in Fig. 6 illustrate that NODEs with recognition models can incorporate gradual priors for learning SSMs from partial and noisy observations. Note that standard methods tailored to the harmonic oscillator may perform better, however, they are not as general nor as flexible.

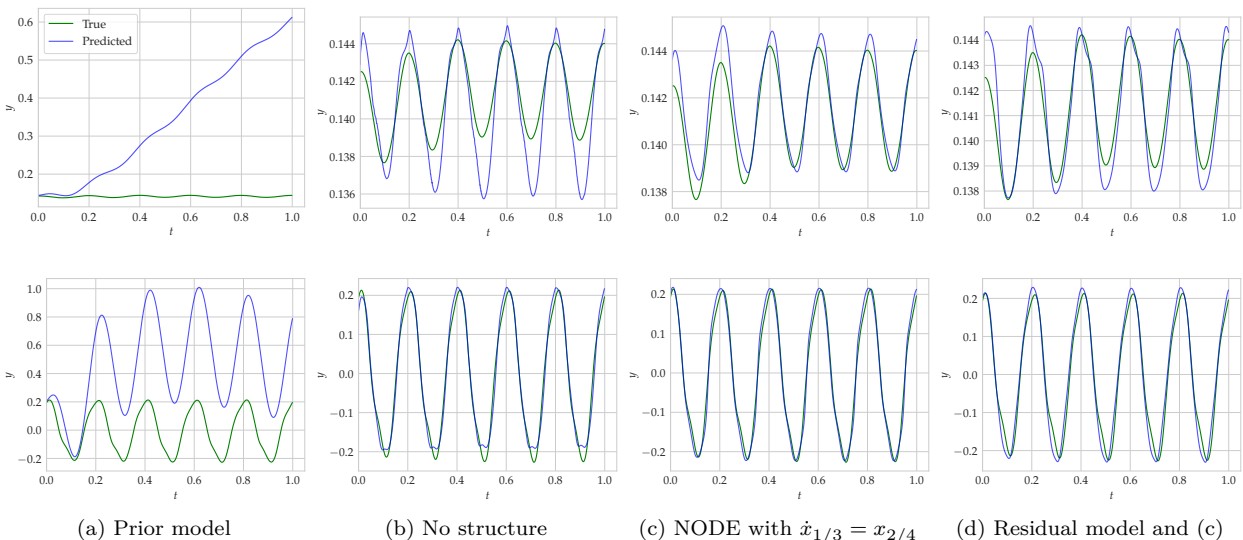

(a) Prior model  (b) No structure  (c) NODE with $\dot{x}_{1/3} = x_{2/4}$  (d) Residual model and (c)

Figure 7: Structured NODEs and KKL recognition on the robotics dataset. After training the NODE on trajectories of 0.2 s from a subset of the input frequencies, we also test on 52 trajectories of 2 s from other input frequencies, to evaluate generalization capabilities (cut at 1 s on plots for visibility). Computing the prediction RMSE for the different structure settings yields: 110 (a), 0.72 (b), 0.66 (c), 1.2 (d). We show one such test trajectory ($x_1$ top row, $x_4$ bottom row) from an unknown initial condition.

## 5.3 Experimental dataset from robotic exoskeleton

We demonstrate the performance of the proposed framework on a real-world dataset. We use a set of measurements collected from a robotic exoskeleton at Wandercraft, presented in (Vigne, 2021) and Fig. 8. This robot features mechanical deformations at weak points of the structure that are neither captured by Computer-Assisted Design modeling nor measured by encoders. These deformations, when measured by a motion capture device, can be shown to account for significant errors in foot placement. Further, they exhibit nonlinear spring-like dynamics that complicate control design. The dataset is obtained by fixing the robot basin to a wall and sending a sinusoidal excitation to the front hip motor at different frequencies. The sagittal hip angle is measured by an encoder, while the angular velocity of the thigh is measured by a gyroscope. In (Vigne, 2021), first results are obtained using linear system identification: the observed deformation is modeled as a linear spring in the hip, and this model is linearized around an equilibrium point, then its parameters are identified. These estimates are sufficient for tuning a robust controller to compensate for the deformation[5]. We aim to learn a more accurate model of this dynamical system of dimension $d_x = 4$, where $y = (x_1, x_4)$, by identifying the nonlinear deformation terms.

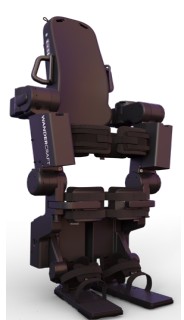

Figure 8: Robotic exoskeleton by Wandercraft.

We investigate three settings: no structure, imposing $\dot{x}_1 = x_2$ and $\dot{x}_3 = x_4$, and learning the residual of the prior linear model on top of this structure as in Yin et al. (2021), by using $f = f_{lin} + f_\theta$ as the dynamics, where $f_{lin}$ is the linear prior. In each setting, we learn from $N = 265$ trajectories of length 0.2 s in a subset of input frequencies. We use a recognition model with $t_c = 0.1$ s. One short test trajectory in the trained frequency regime is shown in the supplementary material and illustrates the data fit. One longer test trajectory with an input frequency outside of the training regime is shown in Fig. 7 and illustrates the generalization capabilities in all three settings and for the linear prior model.

The obtained results demonstrate that structured NODEs with recognition models can identify real-world nonlinear systems from partial and noisy observations. The learned models can fit data from a complex nonlinear system excited with different input frequencies, and somewhat generalize to unseen frequencies.

---

[5]More details are provided in (Vigne, 2021) and in the supplementary material.

| Recognition model | Short rollouts in trained regime | Long rollouts with unseen frequencies | Long rollouts with EKF: $y = (x_1, x_4)$ | Long rollouts with EKF: $y = x_1$ |
|---|---|---|---|---|
| direct | 0.11 | 0.58 | 0.12 | 0.44 |
| RNN+ | 0.15 | 0.56 | 0.15 | 0.37 |
| KKL | 0.17 | 0.60 | 0.12 | 0.37 |
| KKLu | 0.18 | 0.65 | 0.11 | 0.34 |

Table 3: RMSE on test trajectories for the robotics dataset while imposing $\dot{x}_{1/3} = x_{2/4}$. We trained only one model per method; hence, the results only illustrate that all methods are comparable.

The predictions are not perfect, but much more accurate than those of the prior model, as seen in Fig. 17; this is enough to be used in closed-loop tasks such as control or monitoring. Imposing $\dot{x}_{1/3} = x_{2/4}$ leads to similar performance as without structure, but a physically meaningful state-space representation that can be interpreted in terms of position and velocity. Due to the inaccurate predictions of the prior model, learning its residuals leads to lower performance.

With all levels of prior knowledge, the different recognition models lead to comparable results, as illustrated in Table 3. The direct and RNN+ methods lead to slightly lower error on the test rollouts, but also take longer to train due to the higher number of free parameters. The KKL and KKLu methods lead to similar performance. To obtain these results, the choice of the gain matrix $D$ was critical. Rigorous analysis of the role of this parameter is beyond the scope of this article, and remains a relevant task for future work (Buisson-Fenet et al., 2022). We also evaluate the learned model inside an Extended Kalman Filter (EKF). The EKF is a classical state estimation tool for nonlinear systems, that takes the measurement and control as input and outputs a probabilistic estimate of the current underlying state. At each time step, it linearizes the dynamics model and output map then proceeds as a linear Kalman Filter to estimate the mean and covariance of the current state (Krener, 2003). We implement an EKF which receives either $y = h(x) = (x_1, x_4)$ or only $y = h(x) = x_4$ as the measurement, and uses the linear prior or an NODE as dynamics function. In both cases, the NODE estimates are more accurate than those obtained with the linear prior model, as shown in Fig. 9 for a long test trajectory with an input frequency outside of the training regime. As expected, the accuracy for $y = (x_1, x_4)$ is high since we are directly estimating the output. However, the performance difference indicates that the NODE is much more accurate and should enable meaningful state estimation for downstream tasks. When $y = x_1$, the EKF using the prior model is off, while it provides reasonable estimates in most frequency regimes when using the learned model.

## 6 Conclusion

The general formulation of NODEs is well suited for nonlinear system identification. However, learning physically sound dynamics in realistic settings, i.e., with control inputs and partial, noisy observations, remains challenging. To achieve this, recognition models are needed to efficiently link the observations to the latent state. We show that notions from observer theory can be leveraged to construct such models; for example, KKL observers can filter the information contained in the observations to produce an input of fixed dimension for which a suitable recognition model is guaranteed to exist. We propose to combine recognition models and existing methods for physics-aware NODEs to build a unifying framework, which can learn physically interpretable models in realistic settings. We illustrate the performance of KKL-based recognition in numerical simulations, then demonstrate that the proposed end-to-end framework can learn SSMs from partial observations with an experimental robotics dataset. While these observer-based recognition models are demonstrated in the context of NODEs, they are a separate contribution, which can also be used in various system identification methods; they could be combined with e.g., Neural Controlled Differential Equations (Kidger et al., 2020), Bayesian extensions of NODEs (Norcliffe et al., 2021a; Yildiz et al., 2019) or in general optimization-based system identification methods (Schittkowski, 2002; Villaverde et al., 2021).

The results herein illustrate that observer theory and, in particular, KKL observers are suitable for building recognition models. To the best of our knowledge, this work is the first to propose this connection, hence, it

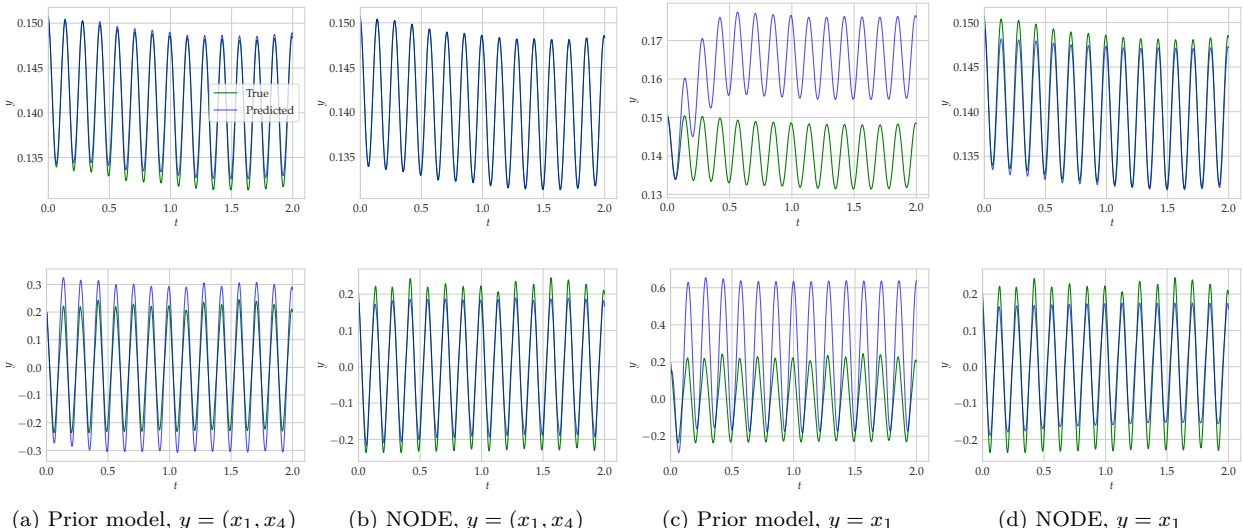

(a) Prior model, $y = (x_1, x_4)$    (b) NODE, $y = (x_1, x_4)$    (c) Prior model, $y = x_1$    (d) NODE, $y = x_1$

Figure 9: State estimation with an EKF on the robotics dataset. After training the NODE with KKL recognition while imposing $\dot{x}_{1/3} = x_{2/4}$, we run the EKF on long test trajectories from unseen input frequencies. When $y = (x_1, x_4)$, both prior and learned model are able to reconstitute the output ($x_1$ top row, $x_4$ bottom row), but the EKF with NODE performs better. When measuring only $x_1$, the EKF using the prior model is off, while its estimates are reasonable in most frequency regimes with the learned model.

also points to remaining open questions. In particular, the choice of $(D, F)$ plays a role in the performance of KKL observers (Buisson-Fenet et al., 2022), and methods for tuning them are still needed; setting $D$ to a HiPPO matrix Gu et al. (2021; 2022) could be an interesting first step. On another note, NODEs can be combined with Convolutional Neural Networks to capture spatial dependencies and learn Partial Differential Equations (PDEs), as investigated in Dulny et al. (2021); Xu et al. (2021). In this paper, we only consider dynamical systems that can be modeled by ODEs, but expect the proposed approach to extend to PDEs.

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

# A    More detailed background on KKL observers

We recall the main existence results on KKL observers. Correctly stating these results requires more formalism than what is used in the main body of the paper. However, the assumptions and the reasoning are identical. We start with the main existence result on autonomous systems, then recall the extension to nonautonomous systems.

## A.1    Autonomous systems

Consider the following autonomous nonlinear dynamical system

$$\dot{x} = f(x), \qquad y = h(x) \tag{10}$$

where $x \in \mathbb{R}^{d_x}$ is the state, $y \in \mathbb{R}^{d_y}$ is the measured output, $f$ is a $C^1$ function and $h$ is a continuous function. The goal of observer design is to compute an estimate of the state $x(t)$ using the past values of the output $y(s)$, $0 \le s \le t$. We make the following assumptions:

**Assumption 1** *There exists a compact set $\mathcal{X}$ such that for any solution $x$ of (10), $x(t) \in \mathcal{X}$ $\forall t \ge 0$.*

**Assumption 2** *There exists an open bounded set $\mathcal{O}$ containing $\mathcal{X}$ such that (10) is backward $\mathcal{O}$-distinguishable on $\mathcal{X}$, i.e., for any trajectories $x_a$ and $x_b$ of (10), there exists $\bar{t} > 0$ such that for any $t \geq \bar{t}$ such that $(x_a(t), x_b(t)) \in \mathcal{X} \times \mathcal{X}$ and $x_a(t) \neq x_b(t)$, there exists $s \in [t - \bar{t}, t]$ such that*

$$h(x_a(s)) \neq h(x_b(s))$$

*and $(x_a(\tau), x_b(\tau)) \in \mathcal{O} \times \mathcal{O}$ for all $\tau \in [s, t]$. In other words, their respective outputs become different in backward finite time before leaving $\mathcal{O}$.*

This is the assumption of backward distinguishability, i.e., Definition 1 but in backward time. It means that the current state is uniquely determined by the past values of the output. On the contrary, (forward) distinguishability means that the initial state is uniquely determined by the future values of the output. If the solutions of (10) are unique, e.g., if $f$ is $C^1$, then these two notions are equivalent.

The following Theorem derived in Andrieu & Praly (2006) proves the existence of a KKL observer.

**Theorem 1 ((Andrieu & Praly, 2006))** *Suppose Assumptions 1 and 2 hold. Define $d_z = d_y(d_x + 1)$. Then, there exists $\ell > 0$ and a set $S$ of zero measure in $\mathbb{C}^{d_z}$ such that for any matrix $D \in \mathbb{R}^{d_z \times d_z}$ with eigenvalues $(\lambda_1, \ldots, \lambda_{d_z})$ in $\mathbb{C}^{d_z} \setminus S$ with $\operatorname{Re} \lambda_i < -\ell$, and any $F \in \mathbb{R}^{d_z \times d_x}$ such that $(D, F)$ is controllable, there exists an injective mapping $\mathcal{T} : \mathbb{R}^{d_x} \to \mathbb{R}^{d_z}$ that satisfies the following equation on $\mathcal{X}$*

$$\frac{\partial \mathcal{T}}{\partial x}(x) f(x) = D\mathcal{T}(x) + Fh(x), \tag{11}$$

*and its left inverse $\mathcal{T}^* : \mathbb{R}^{d_z} \to \mathbb{R}^{d_x}$ such that the trajectories of (10) remaining in $\mathcal{X}$ and any trajectory of*

$$\dot{z} = Dz + Fy \tag{12}$$

*verify*

$$|z(t) - \mathcal{T}(x(t))| \leq M |z(0) - \mathcal{T}(x(0))| e^{-\lambda_{\min} t} \tag{13}$$

*for some $M > 0$ and with*

$$\lambda_{\min} = \min \left\{ |\operatorname{Re} \lambda_1|, \ldots, |\operatorname{Re} \lambda_{d_z}| \right\}. \tag{14}$$

*This yields*

$$\lim_{t \to +\infty} |x(t) - \mathcal{T}^*(z(t))| = 0. \tag{15}$$

## A.2 Nonautonomous systems

These results are extended to nonautonomous systems in Bernard & Andrieu (2019). The system equations are then

$$\dot{x} = f(x, u), \qquad y = h(x, u) \tag{16}$$

where $u \in \mathcal{U}$ is the input. Assumption 2 naturally extends to nonautonomous systems if it is true for any fixed input $u$ of interest. The following Theorem proves the existence of a KKL observer in the nonautonomous case under the weak assumption of backward distinguishability.

**Theorem 2 ((Bernard & Andrieu, 2019))** *Take some fixed input $u \in \mathcal{U}$. Suppose Assumptions 1 and 2 hold for this $u$ with a certain $\bar{t}_u \geq 0$. Define $d_z = d_y(d_x + 1)$. Then, there exists a set $S$ of zero measure in $\mathbb{C}^{d_z}$ such that for any matrix $D \in \mathbb{R}^{d_z \times d_z}$ with eigenvalues $(\lambda_1, \ldots, \lambda_{d_z})$ in $\mathbb{C}^{d_z} \setminus S$ with $\operatorname{Re} \lambda_i < 0$, and any $F \in \mathbb{R}^{d_z \times d_x}$ such that $(D, F)$ is controllable, there exists a mapping $\mathcal{T}_u : \mathbb{R} \times \mathbb{R}^{d_x} \to \mathbb{R}^{d_z}$ that satisfies the following equation on $\mathcal{X}$*

$$\frac{\partial \mathcal{T}_u}{\partial x}(t, x) f(x, u(t)) + \frac{\partial \mathcal{T}_u}{\partial t}(x) = D\mathcal{T}_u(t, x) + Fh(x, u(t)), \tag{17}$$

*and a mapping $\mathcal{T}_u^* : \mathbb{R} \times \mathbb{R}^{d_z} \to \mathbb{R}^{d_x}$ such that $\mathcal{T}_u(t, \cdot)$ and $\mathcal{T}_u^*(t, \cdot)$ only depend on the past values of $u$ on $[0, t]$, and $\mathcal{T}_u(t, \cdot)$ is injective $\forall\, t \geq \bar{t}_u$ with a left-inverse $\mathcal{T}_u^*(t, \cdot)$ on $\mathcal{X}$. Then, the trajectories of (16) remaining in $\mathcal{X}$ and any trajectory of*

$$\dot{z} = Dz + Fy, \tag{18}$$

*verify*

$$|z(t) - \mathcal{T}_u(t, x(t))| \leq M\, |z(0) - \mathcal{T}_u(0, x(0))|\, e^{-\lambda_{\min} t} \tag{19}$$

*for some $M > 0$ and with*

$$\lambda_{\min} = \min\left\{ |\operatorname{Re}\lambda_1|, \ldots, |\operatorname{Re}\lambda_{d_z}| \right\}. \tag{20}$$

*This yields*

$$\lim_{t \to +\infty} |x(t) - \mathcal{T}_u^*(t, z(t))| = 0. \tag{21}$$

**Remark 2** *The literature on nonlinear observer design is wide and many types of observers have been proposed; see Bernard (2019); Bernard et al. (2022) for an overview. The proposed recognition models are based on KKL observers, well-suited for the recognition problem due to the existence of the transformation $\mathcal{T}^*$ which can be approximated jointly with the dynamics. Other designs for general nonlinear systems include high-gain observers, which could lead to the same type of structure for the recognition problem. However, they are subject to peaking and tend to amplify measurement noise. An extended Kalman filter could also be used to directly estimate the trajectories using the current dynamics model. However, there is no convergence guarantee for such filters and they directly rely on a linearization of the current dynamics, hence they are sensitive to model error. We expect this to be harder to train than the KKL-based methods, which decouple recognition and dynamics models.*

## B  Implementation details

We demonstrate the proposed method in several numerical experiments and one experimental dataset obtained on a robotic exoskeleton. We investigate the direct method $\bar{z}(t_c) = (\underline{y}_{0:t_c}, u_{0:t_c})$, the RNN+ method where $\bar{z}(t_c)$ is the output of an RNN run over $(\underline{y}_{0:t_c}, u_{0:t_c})$, the KKL method $\bar{z}(t_c) = (z(t_c), u_{0:t_c})$ and the functional KKL method (denoted KKLu) $\bar{z}(t_c) = z(t_c)$ with $d_z = (d_y + d_u)(d_x + d_\omega + 1)$. The KKL observer is run backward in time: we solve the ODE on $z$ backward in time on $[t_c, 0]$ and learn the mapping from $z(0)$ to $x(0)$, then use all samples to train the NODE model. Similarly for the RNN.

In all cases, the choice of $D$ is important for the KKL-based recognition models. For each considered system and for each considered method, we optimize $D$ once jointly with all other parameters once, then reuse the obtained value of $D$ for all corresponding experiments. We set $F = \mathbb{1}_{d_z \times d_y}$ and initialize $D$ with the following method. We compute the poles $p_i$ of a Butterworth filter of order $d_z$ and cut-off frequency $2\pi\omega_c$ and set each block of $D$ as

$$D_i = \begin{cases} p_i & \text{if } p_i \text{ is real} \\ \begin{pmatrix} \operatorname{Re}\{p_i\} & \operatorname{Im}\{p_i\} \\ -\operatorname{Im}\{p_i\} & \operatorname{Re}\{p_i\} \end{pmatrix} & \text{otherwise} \end{cases} \tag{22}$$

such that $D$ is a block-diagonal matrix, and its eigenvalues are the poles of the filter. This choice ensures that the pair $(D, F)$ is controllable and that $D$ is Hurwitz and has physically meaningful eigenvalues. Other possibilities exist, such as choosing $D$ in companion form, as a negative diagonal... However, we found that this strategy leads to the best performance for the considered use cases. We pick $\omega_c = 1$ for the systems of the recognition model benchmark and the harmonic oscillator with unknown frequency. For the experimental dataset, we initialize as $D = \operatorname{diag}(-1, \ldots, -d_z)$. However, this choice is somewhat arbitrary, and the previous method with $\omega_c = 10$ had similar performance. Principled methods for setting $(D, F)$ are still needed for easing the practical use of KKL observers; setting $D$ to a HiPPO matrix Gu et al. (2022; 2021) could be an interesting first step.

Note also that there is a large part of randomness in the different experiments we present. Hence, results may vary, and obtaining consistent results to rigorously compare the different methods without any statistical variations would require a large computational overhead.

### B.1 Benchmark of recognition models

We demonstrate on numerical systems that an NN-based recognition model can estimate the initial state of a dynamical system from partial measurements. For reproducibility, we generate the training data from simulation and choose systems that can be tested again with reasonable computational overhead.

**Earthquake model**   A simplified model of the effects of an earthquake on a two-story building is presented in Winkel (2017), and an NODE is trained for it in Karlsson & Svanström (2019). This linear model can be written as

$$
\begin{aligned}
\dot{x}_1 &= x_2 \\
\dot{x}_2 &= \frac{k}{m}(x_3 - 2x_1) - F_0\omega^2\cos(\omega t) \\
\dot{x}_3 &= x_4 \\
\dot{x}_4 &= \frac{k}{m}(x_1 - x_3) - F_0\omega^2\cos(\omega t) \\
y &= x_1 + \epsilon,
\end{aligned}
\tag{23}
$$

where $x_1$ and $x_3$ are the positions of the first and second floor respectively, $x_2$ and $x_4$ their velocities, $F_0\omega^2\cos(\omega t)$ is the perturbation caused by the earthquake and only $x_1$ is observed with Gaussian noise of variance $\sigma_\epsilon^2 = 10^{-4}$. We consider the oscillation caused by the earthquake as a disturbance, which is known when simulating training trajectories and unknown to the recognition model: we estimate $x(0)$ from $y_{0:t_c}$ only.

We aim to learn a recognition model that estimates $x(0)$ using only $y_{0:t_c}$ with the methods described above. We set $t_c = 40 \times \Delta t = 40 \times 0.03 = 1.2\,\text{s}$ which seems to be enough to reconstitute the initial condition (after trial and error), $N = 50$ (each sample corresponds to a random initial condition, random $F_0$ and random $\omega$), $n = 100$, and design $\psi_\theta$ (and eventually $f_\theta$) as a fully connected feed-forward network, i.e., a multi-layer perceptron, with two hidden layers containing 50 neurons each, and two fully connected input and output layers with bias terms. The RNN+ model is set to have the same internal dimension $d_z$ as the KKL model. We notice that $t_c$ large enough and enough parameters in $\psi_\theta$, i.e., enough flexibility of the model, are needed for good generalization performance. Also, we pick the sampling time $\Delta t = 0.03\,\text{s}$ low enough such that the obtained trajectories are reasonably smooth, otherwise analyzing the results quantitatively becomes hard due to interpolation errors getting too large.

We train the recognition model with each proposed method and evaluate the results on 100 test trajectories of random initial conditions and random input oscillation. The results on one such test trajectory are illustrated in Figure 10.

We train two settings: either learning a full NODE model (main body of the paper), or having a known dynamics model in which only $k/m$, the main parameter of the dynamics model, is optimized jointly with the recognition model. In our example, we have $k/m = 10$, but we initialize its estimate to a random value in $[8, 12]$. As usual, this problem is not well-posed and there are many local optima. Therefore, we can only hope to converge to a good estimate by starting from a reasonable guess of the main parameter. We keep everything else fixed, including the optimization routine, which might not be the best choice as it has been shown that for parametric optimization, trust region optimization routines with multiple starts often lead to better results (Raue et al., 2013). For the full NODE model, we evaluate the different recognition models by computing the RMSE on the prediction of the output only, since the coordinate system for $x(t)$ is not fixed and a different coordinate system is found in each experiment. For the parametric model, we evaluate the different recognition models by computing the RMSE on the estimation of the whole trajectory over all test scenarios, since the coordinate system for $x(t)$ is fixed by the parametric model. The results are shown in Figure 11 and in the main body of the paper. We observe that the KKL-based models achieve higher

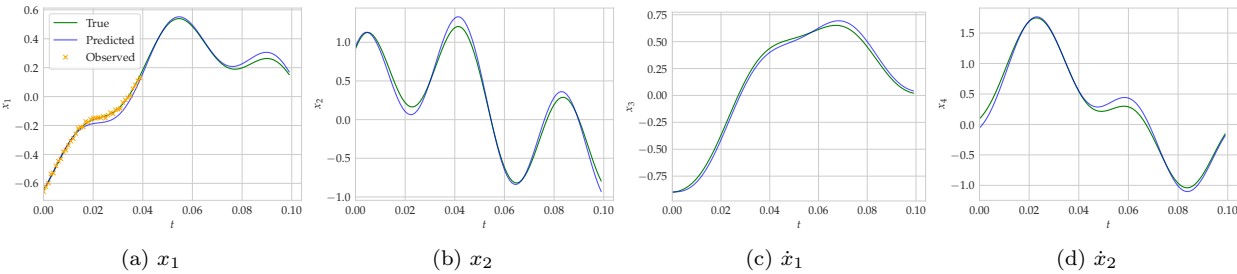

(a) $x_1$        (b) $x_2$        (c) $\dot{x}_1$        (d) $\dot{x}_2$

Figure 10: Test trajectory of the parametric earthquake model with KKL recognition: the initial condition is estimated from $\underline{y}_{0:t_c}$ jointly with the model parameters.

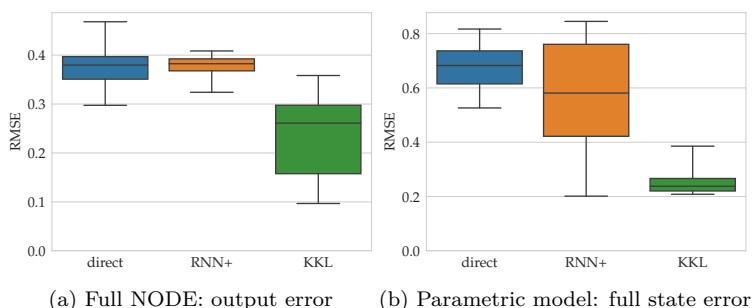

(a) Full NODE: output error      (b) Parametric model: full state error

Figure 11: Results of the obtained earthquake recognition models. We show the RMSE on the prediction of the output when a full NODE model is learned (left column) and of the whole test trajectories when a parametric model is learned (right column). Ten recognition models were trained with the methods direct (left), RNN+ (middle) and KKL (right). The direct method with $t_c = 0$ is not shown here for scaling, but the mean RMSE is over 0.6.

performance, which seems to indicate that the optimization problem based on $z(0)$ is better conditioned than that based on $y_{0:t_c}$.

**FitzHugh-Nagumo model** This model represents a relaxation oscillation in an excitable system. It is a simplified representation of the behavior of a spiking neuron: an external stimulus is received, leading to a short, nonlinear increase of membrane voltage then a slower, linear recovery channel mimicking the opening and closing of ion channels (Clairon & Samson, 2020). The dynamics are written as

$$\dot{v} = \frac{1}{\epsilon}(v - v^3 - u) + I_{ext}$$
$$\dot{u} = \gamma v - u + \beta$$
$$y = v + \epsilon, \tag{24}$$

where $v$ is the membrane potential, $u$ is the value of the recovery channel, $I_{ext}$ is the value of the external stimulus (here a constant), $\epsilon = 0.1$ is a time scale parameter, and $\gamma = 1.5$, $\beta = 0.8$ are kinetic parameters. Only $v$ is measured, corrupted by Gaussian measurement noise $\epsilon$ of variance $\sigma_\epsilon^2 = 5 \times 10^{-4}$.

Our aim is to learn a recognition model that estimates $(v(0), u(0))$ using $y_{0:t_c}$ and $I_{ext}$ with the methods described above. We set $t_c = 40 \times \Delta t = 40 \times 0.03 = 1.2\,\text{s}$, $N = 50$ for 50 random initial conditions and external stimulus, $n = 100$, and design $\psi_\theta$ (and eventually $f_\theta$) as a fully connected feed-forward network, i.e., a multi-layer perceptron, with two hidden layers containing 50 neurons each, and two fully connected input and output layers with bias terms. The RNN+ model is set to have the same internal dimension $d_z$ as the KKL model.

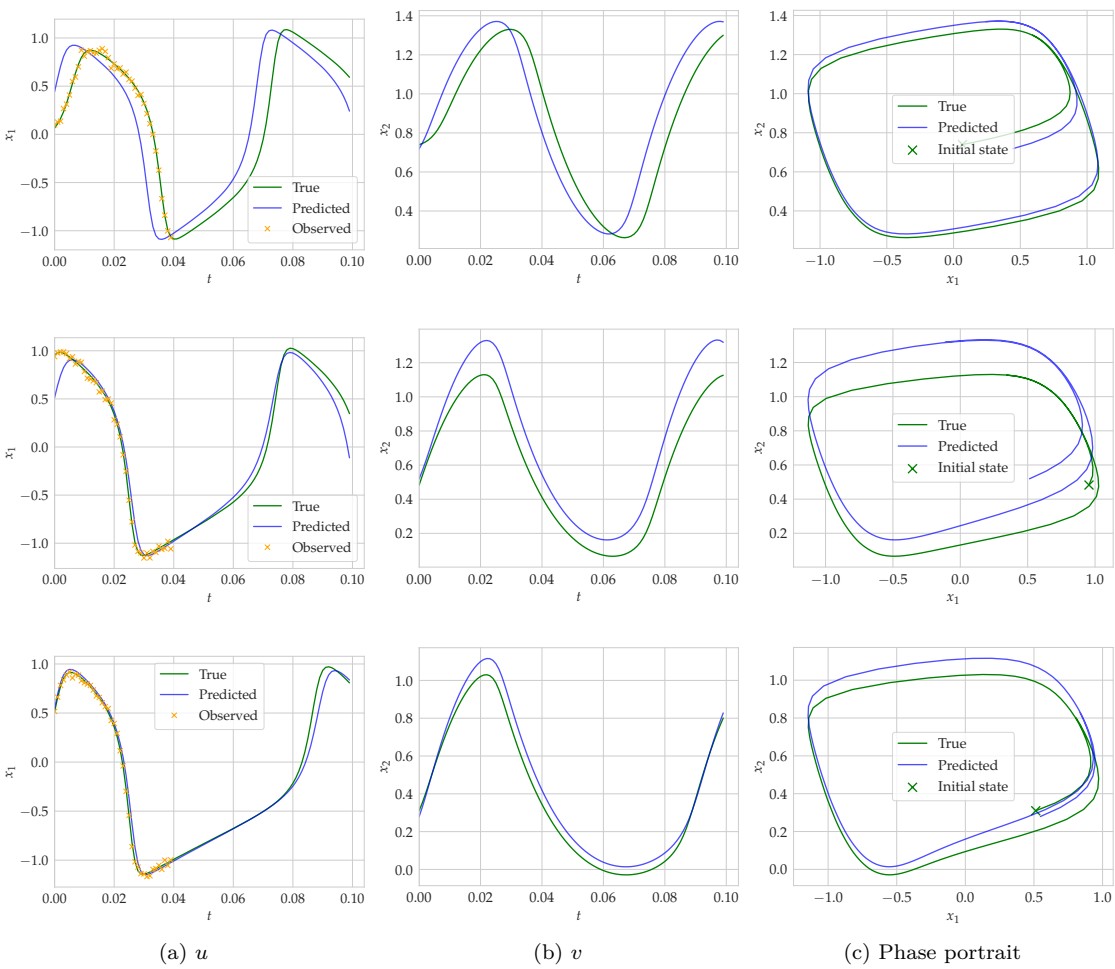

Figure 12: Test trajectory of the parametric FitzHugh-Nagumo model: the initial condition is estimated from $\underline{y}_{0:t_c}$ jointly with the model parameters. We use direct (top), RNN+ (middle) and KKL (bottom) recognition, on three random but similar test trajectories.

We train the recognition model with each proposed method and evaluate the results on 100 test trajectories with random initial conditions and random stimulus. The results on one such test trajectory are illustrated in Figure 12.

We either learn a full NODE model (main body of the paper) or a parametric model for which we estimate the main dynamic parameters $\epsilon$, $\beta$ and $\gamma$ jointly with the recognition model, initialized randomly in $[0.05, 0.15]$, $[0.75, 2.25]$ and $[0.4, 1.2]$ respectively. We evaluate the different recognition models as above. The results are illustrated in Figure 13 and in the main body of the paper. We observe once again that the KKL-based methods lead to lower error.

**Van der Pol oscillator**  Consider the nonlinear Van der Pol oscillator of dynamics

$$\dot{x}_1 = x_2$$
$$\dot{x}_2 = \mu(1 - x_1^2)x_2 - x_1 + u$$
$$y = x_1 + \epsilon, \tag{25}$$

where $x_1$, $x_2$ are the states, $u = 1.2\sin(\omega t)$ is a sinusoidal control input, and $\mu = 1$ is a damping parameter. Only $x_1$ is measured, corrupted by Gaussian measurement noise $\epsilon$ of variance $\sigma_\epsilon^2 = 10^{-3}$.

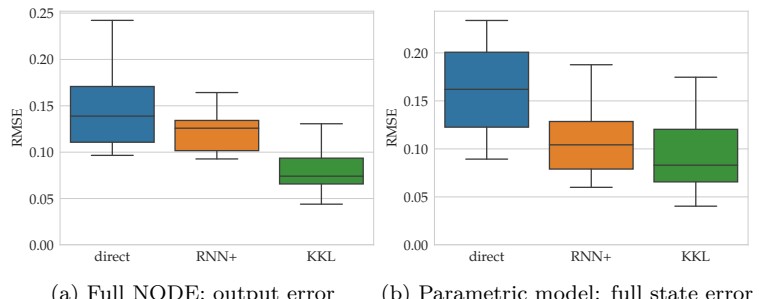

(a) Full NODE: output error      (b) Parametric model: full state error

Figure 13: Results of the obtained FitzHugh-Nagumo recognition models. We show the RMSE on the prediction of the output when a full NODE model is learned (left column) and of the whole test trajectories when a parametric model is learned (right column). Ten recognition models were trained with the methods direct (left), RNN+ (middle) and KKL (right). The direct method with $t_c = 0$ is not shown here for scaling, but the mean RMSE is over 0.4.

Our aim is to learn a recognition model that estimates $x(0)$ using $y_{0:t_c}$ and $u_{0:t_c}$ with the methods described above. We set $t_c = 40 \times \Delta t = 40 \times 0.03 = 1.2\,\text{s}$, $N = 50$ for 50 random initial conditions and values of $\omega$, $n = 100$, and design $\psi_\theta$ (and eventually $f_\theta$) as a fully connected feed-forward network, i.e., a multi-layer perceptron, with two hidden layers containing 50 neurons each, and two fully connected input and output layers with bias terms. Since $u$ is a sinusoidal control input of varying frequency, it can be generated by

$$\begin{aligned}
\dot{\omega}_1 &= \omega_2, \\
\dot{\omega}_2 &= -\omega_3 \omega_1, \\
\dot{\omega}_3 &= 0
\end{aligned} \tag{26}$$

and $u = \omega_1$, where $\omega_1$, $\omega_2$ are the internal states of the sinusoidal system and $\omega_3 > 0$ is its frequency. Hence, we can use the KKLu recognition model with $d_\omega = 3$. The RNN+ model is set to have the same internal dimension $d_z$ as the KKLu model.

We train the recognition model with each proposed method and evaluate the results on 100 test trajectories with random initial conditions and random input frequency. The results on one such test trajectory are illustrated in Figure 14. We also consider either a full NODE model, or a parametric model for which $\mu$ is jointly estimated, after from a random initial guess in $[0.5, 1.5]$. The corresponding box plots are shown in Figure 15. We observe that in both settings, the performance with the different recognition models is very similar. In the main body of the paper, we show the same plot but with higher noise of variance $\sigma_\epsilon^2 = 0.1$ instead of $\sigma_\epsilon^2 = 10^{-3}$ here. Due to the very similar performance, the hierarchy of the different recognition models slightly varies between the noise levels.

## B.2 Synthetic dataset: harmonic oscillator with unknown frequency

We demonstrate the performance of structured NODEs to learn the dynamics of a harmonic oscillator with unknown frequency from partial observations with varying degrees of structure. We train on $N = 50$ trajectories from 50 random initial states in $[-1, 1]^2$ and frequency $\omega^2 = 1\,\text{Hz}$ (i.e., period 6.3 s), of $n = 50$ time steps each for an overall length of 3 s, corrupted by Gaussian measurement noise of variance $\sigma^2 = 10^{-4}$. We use $t_c = 20 \times \Delta t = 1.2\,\text{s}$ for the recognition model. We optimize the parameters using Adam (Kingma & Ba, 2015) and a learning rate of 0.005.

The obtained results are depicted in Figure 16 with the KKL recognition model. We show the prediction of a random test trajectory with random initial state: $y_{0:t_c}$ is measured for this trajectory, used by the recognition model to estimate $x(0)$. Then, the learned NODE is simulated to predict the whole state trajectory for 500 time steps, i.e., ten times longer than the training time to illustrate the long-term behavior. For the quantitative results presented in the main body of the paper, we predict on test trajectories of 150 time steps, i.e., three times the training time. These make the long-term performance difference due to the degree

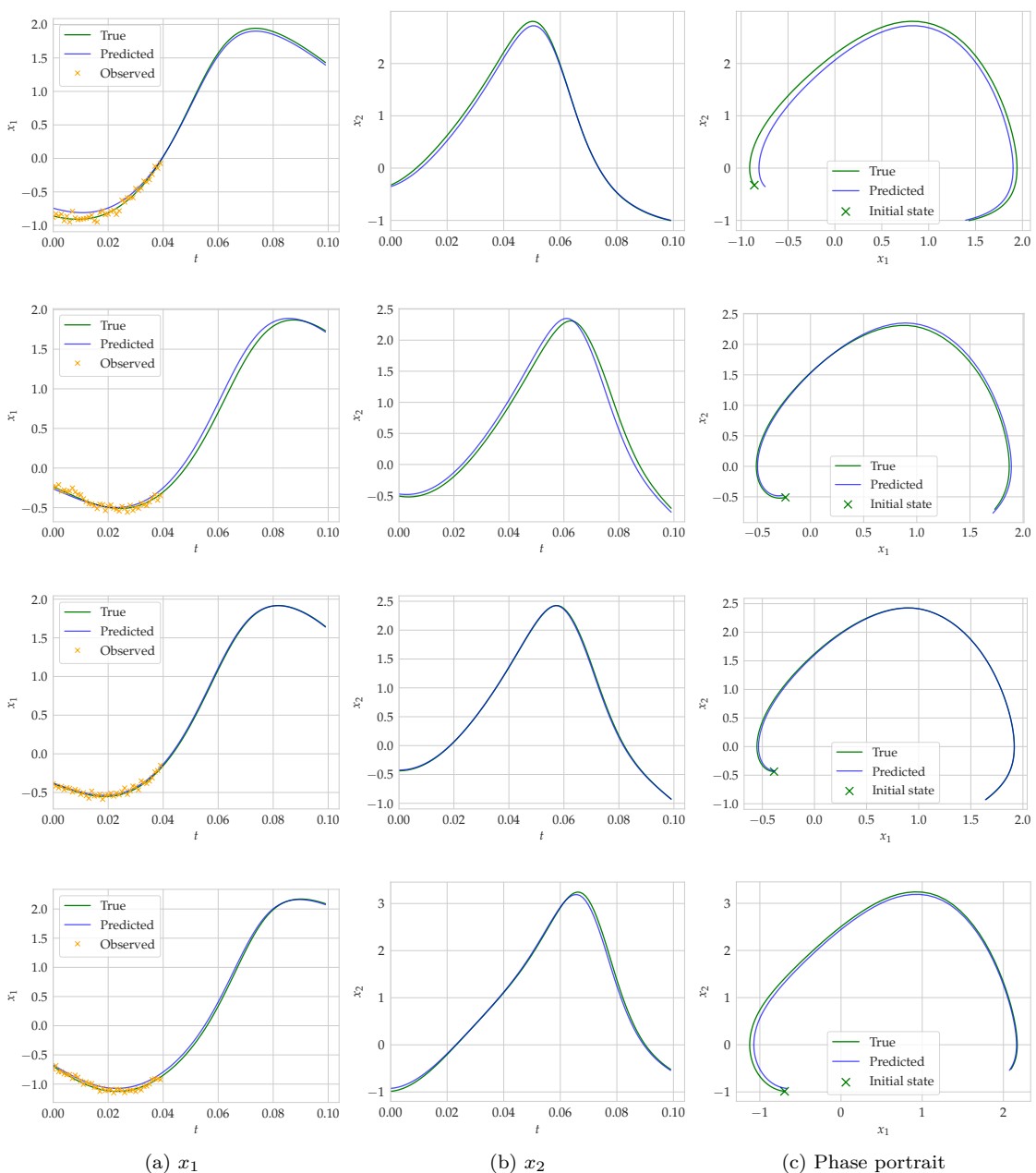

Figure 14: Test trajectory of the parametric Van der Pol model: the initial condition is estimated from $\underline{y}_{0:t_c}$ jointly with the model parameters. We use direct (top), RNN+, KKL and KKLu (bottom) recognition, on four random but similar test trajectories.

of prior knowledge less visible, but lead to more consistent and quantitatively comparable results (with the long test trajectories, the interquartile range of the experiments was very large due to error accumulation which can possibly blow up over a long prediction horizon).

We train the dynamics and recognition model in each setting ten times, for recognition models direct, RNN+ and KKL. The mean RMSE over hundred test trajectories is depicted in the main body of the paper.

**No structure**   We start without imposing any structure, i.e., learning a general latent NODE model of the system as in (4.2) The NODE fits the observations $y = x_1$, but not $x_2$ as it has learned the dynamics in

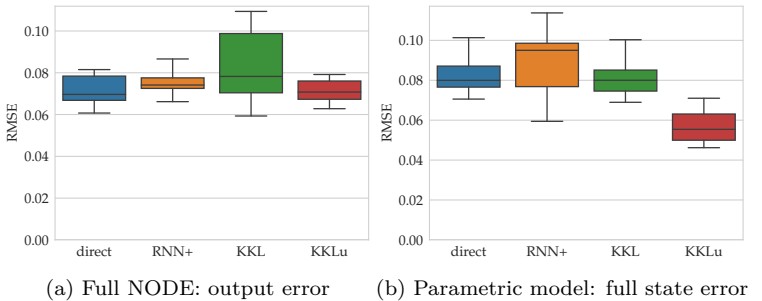

(a) Full NODE: output error          (b) Parametric model: full state error

Figure 15: Results of the obtained Van der Pol recognition models. We show the RMSE on the prediction of the output when a full NODE model is learned (left column) and of the whole test trajectories when a parametric model is learned (right column). Ten recognition models were trained with the methods direct (left), RNN+, KKL and KKLu (right). The direct method with $t_c = 0$ is not shown here for scaling, but the mean RMSE is over 0.6.

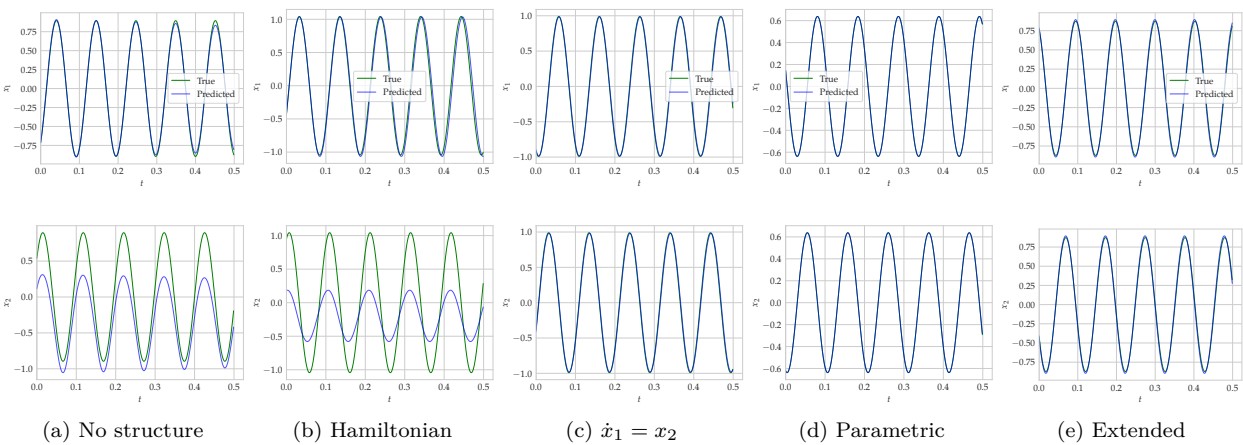

(a) No structure     (b) Hamiltonian     (c) $\dot{x}_1 = x_2$     (d) Parametric     (e) Extended

Figure 16: Random test trajectory of the trained NODE for the harmonic oscillator: without imposing any structure (a), imposing Hamiltonian dynamics (b), imposing $\dot{x}_1 = x_2$ (c), directly identifying a parametric model (d) and learning a recognition model of an extended state-space representation where $x_3 = \omega^2$ (e). We show the true and predicted trajectories of $x_1$ (top) and $x_2$ (bottom).

another coordinate system, which is expected for general latent NODEs. It also does not conserve energy, which is not surprising when no structure is imposed, as discussed e.g., in Greydanus et al. (2019).

**Hamiltonian state-space model**   We now assume the user has some physical insight about the system at hand: it derives from a Hamiltonian function, i.e., there exists $H$ such that

$$\dot{x}_1 = \frac{\partial H}{\partial x_2}(x), \qquad \dot{x}_2 = -\frac{\partial H}{\partial x_1}(x). \tag{27}$$

We approximate $H$ directly with a neural network $H_\theta$ of weights $\theta$, such that the NODE has form (27), and inject this into the optimization problem (4). This formulation enforces the constraint that the dynamics derive from a Hamiltonian function, whose choice is free. In that case, we do not necessarily find the "physical" state-space realization, as several Hamiltonian functions can fit the data. However, the obtained state-space model conserves energy due to the Hamiltonian structure.

**Imposing $\dot{x}_1 = x_2$**   We now impose a somewhat stronger structure in (4):

$$\dot{x}_1 = x_2, \qquad \dot{x}_2 = -\nabla H(x_1), \tag{28}$$

where only the dynamics of $x_2$ need to be learned. This enables the NODE to recover both the initial state and the unknown part of the dynamics in the imposed coordinates while also conserving energy, as this is a particular case of Hamiltonian dynamics with Hamiltonian function $\frac{1}{2}x_2^2 + H(x_1)$.

**Parametric system identification** We now directly learn a parametric model of the harmonic oscillator

$$\dot{x}_1 = x_2, \qquad \dot{x}_2 = -\omega^2 x_1,$$

where $\omega > 0$ is the unknown frequency. We approximate $\omega$ with a parameter $\theta$, which is initialized randomly in $[0.5, 2]$. We obtain excellent results with this method, as $\theta$ is estimated correctly up to $10^{-2}$ and the trained recognition model gives satisfying results. This demonstrates that our framework can recover both the dynamics and the recognition model in the physical coordinates imposed by the parametric model from partial and noisy measurements in this simple use case.

**Extended state-space model** We now consider the extended state-space model

$$\dot{x}_1 = x_2, \qquad \dot{x}_2 = -x_3 x_2, \qquad \dot{x}_3 = 0 \tag{29}$$

where $x_3 = \omega^2$ is a constant state representing the unknwon frequency. In this case, the dynamics are completely known and only the recognition model is left to train, in order to estimate the initial condition $x(0) \in \mathbb{R}^3$, where $x_3(0)$ is the unknown frequency. This is the same degree of structure as the parametric model: the dynamics are known up to the frequency. However, it also is a more open problem: since we learn a recognition model for $x(0) \in \mathbb{R}^3$, at each new trajectory, we estimate a new value of the frequency $x_3(0)$, which was considered the same across all trajectories for the previous methods. Therefore, with this setting, we also obtain models that can predict energy-preserving trajectories in the physical parameters, but with lower accuracy due to this extra degree of freedom.

### B.3 Experimental case study: robotic exoskeleton

We use a set of measurements collected at Wandercraft on one of their exoskeletons and presented in Vigne (2021). More details on the robot, the dataset and the methods applied at Wandercraft are provided in Section 4.1.2.1 of Vigne (2021).

For this experiment, the robot basin is fixed to the wall and low amplitude sinusoidal inputs are sent to the front hip motor with different frequencies between $2\,\text{Hz}$ and $16\,\text{Hz}$. A linear model has been identified in Vigne (2021) by modeling the deformation as a linear spring in the hip motor, yielding a system with $x \in \mathbb{R}^4$. The angle of the hip ($x_1$) is measured by the encoder on this motor, while a gyrometer measures the angular velocity of the thigh ($x_4$). The measurements are sampled with $\Delta t = 1ms$. The aim is to identify the nonlinear deformations in the hip motor, which can be seen in motion capture and cause significant errors in gait planning, but are not captured by the known models of the exoskeleton.

We start by preprocessing the signal: for each input frequency and corresponding trajectory, we compute the FFT of $y$, apply a Gaussian window at $f_c = 50\,\text{Hz}$ on the spectrum, then apply an inverse FFT and slice out the beginning and the end (100 time steps) of each signal to get rid of the border effects. For $u$, which is not very noisy, we rather apply a Butterworth filter of order 2 and cut-off frequency $200\,\text{Hz}$. We cut the long trajectories for each input frequency in slices of 200 samples, and stack these training trajectories of length $0.2\,\text{s}$ together to form our set of training trajectories. Hence, all trajectories have the same sampling times and can be simulated in parallel easily. We choose the length of $0.2\,\text{s}$ because it seems long enough to capture some of the dynamics even in the low-frequency regime, but also short enough to remain acceptable in the high-frequency regime.

We then run the proposed framework on this data. We directly train the NODE on a random subset of training trajectories, use a subset of validation trajectories for early stopping, and a subset of test trajectories to evaluate the performance of the learned model. When no further indications are provided, we use a recognition model of type KKL with $d_z = 10$, $F = \mathbb{1}_{d_z \times d_y}$, which yields 110 for the dimension of $(z(t_c), u_{0:t_c})$. The parameter $D$ was optimized after being initialized at $\text{diag}(-1, \ldots, -10)$. We also trained a recognition model of type KKLu with $d_z = 50$ for which $D$ was also optimized starting at $\text{diag}(-2, \ldots, -100)$, and

recognition models of type direct and RNN+ using the same information contained in $(\underline{y}_{0:t_c}, u_{0:t_c})$ and the same size of the latent state as for KKLu, i.e., dimension 50. Both recognition and dynamics models are feed-forward networks with five hidden layers of 50 respectively 100 neurons and SiLU activation.

We notice that for this complex and nonautonomous use case, the direct and RNN+ recognition methods seem easier to train. However, they also take longer due to having more parameters, and have higher generalizaiton error on the long test rollouts. We also notice that $D$ needs to be chosen well for the KKL-based recognition models to obtain good performance, which needs to be investigated further.

Normalization is also an important aspect in the implementation: all losses and evaluation metrics are scaled to the same range, so that all loss terms play a similar role and remain within a similar range. This ensures that the values on which the optimization is based are always numerically tractable for the chosen solver. Different scaling possibilities are discussed in Sec. 5.2.3 of Schittkowski (2002). In our case, since we do not know in advance the values that $x(t)$ will take, we compute the mean and standard deviation of the samples in $y(t)$ and $u(t)$ and scale all outputs $y(t)$ respectively inputs $u(t)$ according to these. We also scale all states $x(t)$ or derivatives $\dot{x}(t)$ using the scaler on $y(t)$ for the dimensions that are measured ($x_1$ and $x_4$), and a mean of the scaler on $y(t)$ for the other dimensions. This is not quite correct, but it is the best we can do without knowing the range of values that $x(t)$ will take, and it is enough to ensure that all scaled values of $x(t)$ stay within a reasonable range.

We investigate three settings: no structure, imposing $\dot{x}_1 = x_2$ and $\dot{x}_3 = x_4$ ("structural" prior), and learning the residuals from the prior linear model on $\dot{x}_2$ and $\dot{x}_4$ ("regularizing" prior) with $\lambda = 5 \times 10^{-7}$; we already have $\dot{x}_1 = x_2$ and $\dot{x}_3 = x_4$ in the prior model). For evaluating the prior linear model, we use the estimated initial states obtained by the recognition model of the last setting, in order to be in the coordinate system that corresponds to the prior. In each setting, we learn from $N = 265$ trajectories of a subset of input frequencies: $\{2.5, 3.5, 4.5, 5.5, 6.5, 7.5, 8.5, 9.5, 11, 13, 15\}$ Hz. We then evaluate on 163 test trajectories of $0.2\,\mathrm{s}$ from these input frequencies, to evaluate data fitting in the trained regime. We also evaluate on 52 longer ($2\,\mathrm{s}$) test trajectories from other input frequencies, to evaluate the interpolation capabilities of the learned model: $\{2, 3, 4, 5, 6, 7, 8, 9, 10, 12, 15, 17\}$ Hz. We use the Adam Kingma & Ba (2015) optimizer with decaying learning rate starting at $8 \times 10^{-3}$ for the first two settings, $5 \times 10^{-3}$ for the third setting.

The obtained results are described in the main body of the paper. One longer test trajectory with an input frequency outside of the training regime is presented in Fig. 7 when imposing $\dot{x}_1 = x_2$ and $\dot{x}_3 = x_4$. Overall, we find that structured NODEs are able to fit this complex nonlinear dynamical system using real-world data and realistic settings. The predictions of the obtained models are not perfect, but they are much better than those of the prior model, such that they could probably be used in a closed-loop control task like the linear model currently is at Wandercraft. This is confirmed by implementing an EKF that uses the learned models for state estimation (tuning fixed for all recognition models and all degrees of prior knowledge). Adding structure leads to similar performance, but to a model that can be physically interpreted in terms of position and velocity. In the third setting (hard constraints and residuals model), the accuracy is lower. This is due to the fact that the linear prior model leads to rather inaccurate predictions.

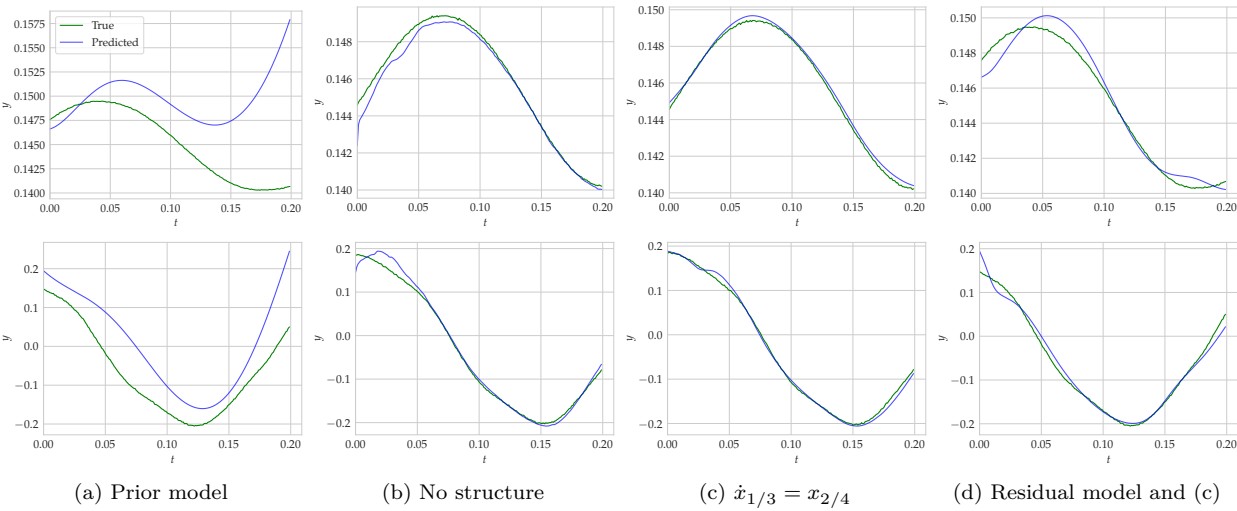

(a) Prior model      (b) No structure      (c) $\dot{x}_{1/3} = x_{2/4}$      (d) Residual model and (c)

Figure 17: Structured NODEs and KKL recognition on the robotics dataset. We test on 163 trajectories of $0.2\,\mathrm{s}$ from the same input frequencies as the training data to evaluate data fitting in the trained regime, and compute the RMSE: we obtain respectively 5.6 (a), 0.16 (b), 0.18 (c), 0.31 (d). We show one such test trajectory ($x_1$ top row, $x_4$ bottom row) from an unknown initial condition.

