# OpenReview forum: "Recognition Models to Learn Dynamics from Partial Observations with Neural ODEs"
_TMLR — Accepted by TMLR_

### Review · Reviewer_Gp37 · 2022-11-07

**Summary Of Contributions:**

The main topic of the paper is the construction of recognition networks for neural ODE's initial state. The authors propose to utilize the theory of the Kazantzis-Kravaris/Luenberger (KKL) observer and to build a recognition network on top of it. The effectiveness of the proposed recognition network is examined empirically.

The authors discuss another topic, the spectrum of the extent of structured-ness of neural ODEs. In some of the experiments, they examine the differences between neural ODEs with different levels of prior knowledge incorpolated.

**Audience:**

Yes

**Claims And Evidence:**

Yes

**Requested Changes:**

### 1.

The interrelation between the two topics of the paper (i.e., recognition models and structured NODEs) should be discussed more clearly. These are interesting yet two independent topics, at least given what I could read from the current manuscript. The proposed method is indeed applied to models with different levels of prior knowledge in Sections 5.2 and 5.3, but it is still unclear why the proposed KKL recognition model is (supposed to be) particularly suitable for such models with gradual priors.

### 2. (related to Point 1)

Please elaborate on the following claim in Section 5.2:

> These results illustrate that NODEs with recognition models can flexibly incorporate
> gradual priors for learning dynamics from partial and noisy observations.

Although I do not disagree with it, the intention of such a claim is ambiguous. Do you mean "NODEs with recognition models" *in general* can flexibly incorporate gradual priors? Or, do you mean the proposed KKL recognition models specifically? If the authors' intention is the former, then, again, it is unclear why the paper discusses this topic of gradual priors together with the topic of KKL recognition models. If the authors' intention is the latter, the current empirical results do not support such a claim.

### 3.

In the proposed method, a sequence of $y$ is first put into the system of $z$ in Eq.(6), which is integrated backward in time. This structure reminded me of "state space layers" [Gu+ 2021; Gu+ 2022]. It would be great if the relationship between them and the proposed recognition models could be discussed.

- [Gu+ 2021] https://openreview.net/forum?id=yWd42CWN3c
- [Gu+ 2022] https://openreview.net/forum?id=uYLFoz1vlAC

---

Below are minor points.

### 4.

In the first half of Section 2.2, the categorization of papers into "soft" and "hard" constraints sounds sometimes strange.

- The stable deep dynamics model by Manek & Kolter (2019) is classified as a "soft" approach, with which I do not agree. Their method ensures stability by construction without any need for regularization.
- The authors define the "hard" approach as something in which constraints are enforced in the optimization problems, but it is not always the case in the papers introduced as examples of the "hard" approach. For example, the Hamiltonian neural nets are usually learned by an unconstrained optimization; there is no need for constraints because the loss function of HNNs encodes the physical knowledge by construction.

### 5.

In Section 4, notations $y([0,t_c])$ and $y_{0:t_c}$ seem mixed. Please unify if they denote the same notion. If not, please specify what each notation means specifically.

### 6.

At the beginning of Section 4.1, the authors say

> Some recognition methods have been proposed in the literature, not necessarily for system identification
> with NODEs, rather for probabilistic (Doerr et al., 2017) or generative Yildiz et al. (2019) models.

And later in the bottom of Page 5, they say

> There are few works on NODE-based system identification from
> partial observations, some of which train a recognition model from $y_{-t_c:0}$ (Ayed et al., 2020; Yildiz et al.,
> 2019; Norcliffe et al., 2021a), or ...

The work of Yildiz et al. (2019) appears in the both places, which is contradictive. In my understanding, the latter is true.

### 7.

In Section 5.3, why did you defer the result on the "longer test trajectory with an input frequency outside of the training regime" to the appendix? I think it would be much more interesting than the rather trivial result that is currently presented in Section 5.3

### 8.

In Section 5.3, the idea and the procedures of the experiment with EKF should be described in more detail.

### 9.

Please verify if the subcaptions of Figure 16 are correct.

**Strengths And Weaknesses:**

### Strengths

- The main contribution of the paper, i.e., the construction of the KKL observer-based recognition networks, is interesting.
- The experiments indeed show the possible utility of the proposed method.

### Weaknesses

- The interrelation between the two topics of the paper (i.e., recognition models and structured NODEs) is unclear.

---

> ### Author Response · Authors · 2022-11-24
> **Replies to review**
>
> We thank the reviewer for their thorough review of our paper.
>
> *Remark: The interrelation between the two topics of the paper (i.e., recognition models and structured NODEs) should be discussed more clearly.*
>
> Our overarching goal is to propose an end-to-end framework for learning physical models from experimental data. We focus on learning state-space models (SSMs), since these are common in system identification and control theory as most modern control synthesis methods build on them, and allow for a physical interpretation.
> For this end-to-end framework, two aspects are needed:
> + A structured SSM: when identifying dynamical systems, physical knowledge is often available, so the chosen SSM learning method should be able to include this knowledge. NODEs are a powerful tool for this, as they leverage the predictive power of deep learning and can  include a wide range of prior knowledge in a flexible manner. There exist many previous works on such approaches, which we denote *structured NODEs*.
> + Recognition models to estimate the unknown initial condition from the partial observations. This is our main technical contribution.
>
> Combining these two aspects enables SSM learning from partial observations. Our experiments illustrate this, by showing the proposed recognition models perform well in combination with different levels of structured NODEs.
> We thank the reviewer for pointing out this lack of clarity, and have reformulated the introduction accordingly.
>
> *Remark: This structure reminded me of "state space layers" [Gu+ 2021; Gu+ 2022].*
>
> We thank the reviewer for pointing out these interesting references.
> We now cite them in the conclusion.
> These linear state space layers (LSSL) are indeed similar to the proposed method, and remind us of DeepKKL (Janny et al, 2021): an output predictor is learned, based on an underlying latent state that evolves linearly and is linked with the output by a nonlinear transformation. The main practical difference is that the underlying linear system is a linear filter of the output in DeepKKL, while it is a linear state-space model with a specific structure for LSSL (the matrix $A$ is a HiPPO matrix).
> In both cases, while these methods are related to the proposed framework, their focus lies on learning an output predictor rather than learning a recognition model that identifies the initial condition in a physically meaningful set of coordinates, in which the dynamics can be learned jointly. However, it would be interesting to combine the proposed HiPPO matrices with KKL-based designs: it could improve the performance of KKL recognition models to use a HiPPO matrix as $D$, since these have a property of continuous-time memorization.
>
> *Remark: In the first half of Section 2.2, the categorization of papers into "soft" and "hard" constraints sounds sometimes strange.*
>
> In light of this remark, we have changed our notation into "regularizing" and "structural" priors. We hope this notation makes the distinction clearer between problems which are regularized to encourage certain aspects, and problems which structurally impose these aspects (often through constraints or a specific form of the optimization problem).
>
> We thank the reviewer for the other comments and have implemented them in the revised version. Changes are indicated in blue.

---

### Review · Reviewer_5jQP · 2022-11-09

**Summary Of Contributions:**

This paper introduces a new recognition model for latent ordinary differential equations (ODEs). The framework relies on nonlinear observer designs, in particular Kazantzis-Kravaris/Luenberger (KKL) observer. Under certain assumptions, the proposed encoder design is able to recover the unknown latent state (in the limit of an infinite sequence). It is exhaustively evaluated on a series of dynamical systems with varying degrees of prior knowledge. It is also compared against the encoder architectures utilized in state-of-the-art methods. The proposed method consistently outperforms alternative constructions when tested on simple simulated dynamics as well as a robotic exoskeleton dataset.

**Audience:**

Yes

**Broader Impact Concerns:**

I feel there is no direct adverse ethical implication of this work.

**Claims And Evidence:**

Yes

**Requested Changes:**

- The text could be slightly modified so that GP-ODEs are also included in the taxonomy. Testing the proposed framework on GP-ODEs would be even better but this is not necessary for the paper to be accepted.
- The terms/concepts in Section 4.2 could be explained in more detail. This is crucial for the understandability of the work and contributions.
- I would be happy if the author can reply to the items listed under "Comments" above.



**Strengths And Weaknesses:**

### Strengths
- Applying a KKL-based recognition model to latent ODEs or partially observed dynamical systems is a very natural idea. To my knowledge, this is the only neural ODE (NODE) encoder based on rigorous theory, which in turn comes with certain guarantees. On its own, this is a valuable contribution.
- The experiments are very elaborate. KKL-based encoder almost always achieves better results or is on par. This is one of the strongest aspects of the work!
- Overall, the paper is written well. In particular, I enjoyed reading the related work.

### Weaknesses
- Certain parts of the text can be improved, see the notes at the end of this section. Since the main audience will be the machine learning community, several terms/concepts in Section 4.2 should be explained in more detail. For instance,
  - What does $z$ denote?
  - What do "Hurwitz" and "controllable pair" mean?
  - What does "observer" mean intuitively?
  - What are the dimensionalities of $D$ and $F$, and why are they so?
- In general, the measurement function $h$ is assumed to be known. I believe this could be slightly too restrictive.
- Figure-1 provides a taxonomy of ODE methods but it completely disregards Gaussian process (GP) based ODE systems (this applies to the related work). I believe a better way of presenting this taxonomy would be "black-box ODEs", "structured ODEs" and "parametric models".


### Comments
I would be happy if we can chat a bit on the following:
- I'm not sure if the word "soft" fits in Figure-1. Approaching from the "additional penalty terms" perspective, it does feel soft. However, including partially known dynamics in the model formulation is a very strong inductive bias and sounds much stronger than "soft".
- _“hard” constraints in the optimization problem_: Don't the methods listed after this phrase undergo significant architecture changes (compared to vanilla NODEs)? I would call this "hard constraints on the (dynamics) formulation / differential function" instead of "optimization problem" as the latter is a much broader concept that includes completely unrelated topics, e.g., combinatorial/constrained/discrete optimization.
- _We can then learn a residual NODE on top_: This is vague and requires a more detailed explanation.
- $y^j(t_i)$ reads as if the data trajectory is continuous. I suggest using $y^j_i$.
- About remark-1: Extremely expressive h would surely make identification difficult. I believe this remark requires elaboration.
- _for $t_c$ large enough_: How would one quantify that?
- _y is the output of (1)_: Looking at (1), the output $y$ depends on the initial value $x_0$. Then, KKL-based recognition model, which generates $x_0$, depends on $y$. Isn't this a circular dependency?
- _The direct method with $t_c = 0$ was run but_: I'm not sure if this is reasonable/fair. Do other methods take only the initial observation as input? If so, does RNN+ make sense at all?
- _the more the observer has converged, the closer the relationship_: That is true; however, do we expect $x(0)$ and $T^*(z(0))$ to be the same up to the added Gaussian noise?
- Harmonic oscillator experiment is super nice!
- It would be nice to understand more about why Figure-7d is worse compared to 7b. Why the residual model cannot correct the linear dynamics?
- Can the model be applied to more complex datasets, i.e., images? If so, how? If not, why not?

### Minor comments
These are very minor comments, feel free to disregard them if you don't think they fit:
- _The behavior of such systems often follows a certain structure, which can be inferred, for example from the laws of physics_: Does the behavior or the system follow a certain structure? Here, does "inferred" refer to a human being observing a system and making conclusions concerning the structure or inference from data using ML tools?
- _even an accurate model cannot account for all aspects_: all aspects of what?
- I'm not sure if "this experimental data" or "this data" are grammatically correct.
- _In general, learning the dynamics is an ill-posed problem_: Including references would be nice.
- _Learning latent representations of the available data_: Maybe replace with "Learning latent representations from data" as there is no "non-available data"
- _a continuous-time ODE_: Is there a "discrete-time ODE"?
- _However, as for most machine learning methods, little insight into the desired latent representation_: Up until here, the reader is not informed about latent NODEs. In fact, the sentence "_The aim is to approximate a vector field_" feels like NODEs are defined solely in the data space.
- _However, in the case of partial observations, the unknown latent state_: Again, why/when NODEs are latent should be discussed before this.
- _We show that the KKL-based recognition models perform well_: This can be elaborated just a bit; why do they perform better, what is the performance metric, etc.
- _We formulate NODEs as_: Maybe re-formulate as it is already formulated and is flexible? The novel bit in this "formulation" should be highlighted.
- _We introduce recognition models to link observations and latent state_: This was already done in Chen'18. Maybe say something like "new recognition models based on KKL to link..."?
- _In the area of system theory_: In system theory
- _The topic of system identification aims at_: System identification aims at
- _Observers often assume a good dynamics model_: What is meant by "good models"?
-  _In that case, the unknown parts_: In which case; former or latter?
- _can ensure certain physical properties_: Maybe a few examples?
- _do not verify important physical principles such as conservation law_: Maybe satisfy or infer instead of verify?
- _for an overview of this approach_: Maybe these efforts?
- _the take on recognition models_: This sounds a bit too informal.
- _Initial conditions $x_a$_: Maybe $x_{a,0}$ to stress initial?
- _as a neural network that takes as input $z(t_c)$_: At least I felt like "where does this term come from?"
- Only after reading Section 4.1, I understood that "direct methods" rely on stacking whereas recurrent methods rely on dynamical encoders. I suggest mentioning this earlier.
- The whole Section 4.1 could be a separate "background" section. It seems to disrupt the natural flow of "problem definition, methodology, experiments".
- _to the latent state_: I would say to the latent *initial* state.
- In Figure-2 caption, I would say _the unknown parameters of the *known osciallating* dynamical systems_.
- trajectories of "3 seconds" as it is not clear at first.
- I would fix the y-axis scale in Figure-3.
- I would match (a)-(e) in Figure-5 with Figure-1.
- _learning the residual of the prior linear model on top of this structure_: This requires more explanation: Do you use an additive differential function?
- The performance metric in Table 3 should be noted.
- Maybe a very brief summary of results in the conclusion, e.g., in 2 sentences?

---

> ### Author Response · Authors · 2022-11-24
> **Replies to review**
>
> We thank the reviewer for the positive evaluation of our paper.
>
> *Remark: Since the main audience will be the machine learning community, several terms/concepts in Section 4.2 should be explained in more detail.*
>
> We thank the reviewer for pointing this out. We have expanded Section 4.2 to make the concepts involved in KKL observers clearer.
>
> *Remark: In general, the measurement function $h$ is assumed to be known. I believe this could be slightly too restrictive.*
>
> The output map $h$ is assumed known in general. This is often the case in a system identification setting, i.e. when dealing with a dynamical system with a physically meaningful state-space form, because the user knows which states or compositions of states are being measured. However, if that is not the case, an expressive output map can be learned jointly with the dynamics and recognition models, as is done in general latent NODEs (Rubanova et al, 2019; Yildiz et al, 2019; Norcliffe et al, 2021a). We agree that this, however, complicates the identification task.
>
> *Remark: Figure-1 provides a taxonomy of ODE methods but it completely disregards Gaussian process (GP) based ODE systems (this applies to the related work). I believe a better way of presenting this taxonomy would be "black-box ODEs", "structured ODEs" and "parametric models".*
>
> We are aware of the literature on GP-ODEs, however, we have tried to restrict the related work mainly to NODEs for brevity. We do mention GP-ODEs in the related work on learning physics-aware models (Section 2.2), and when introducing direct recognition models (Section 4.1). We agree with the reviewer that it would be more general to include them also in our taxonomy in Fig. 1, and have modified it accordingly. In this paper, we focus on NODEs because their flexible formulation can easily include a large range of physical priors, and they can deal with large datasets.
> It would indeed be interesting to compare a probabilistic version of KKL recognition models to standard direct recognition in combination with GP-SSM instead of NODEs, but we leave this for future work.
>
> Comments:
> + We have replaced "soft" with "regularizing" and "hard" with "structural" priors.
> + We mean that we can learn the residuals of the prior model as a vanilla NODE, by using $f_{prior}(x,u) + f_\theta(x,u)$ as the dynamics model while (slightly) penalizing $\left| f_\theta(x,u) \right|$ as in (Yin et al, 2021). We have clarified.
> + How long $t_c$ must be is a difficult question: the longer $t_c$, the more observable we expect the system to be and the more the observer has converged, hence the easier the identification should be. We show in the ablation study (Fig. 3) that KKL recognition reaches a plateau once $t_c$ is "large enough". However, analyzing in a principled way for which minimal value of $t_c$ a given nonlinear system is observable remains an open question in control theory.
> + We apologize for the lack of clarity: the signal $y$ in the KKL observer (6) and (8) is the measurement, hence it is actually $\underline{y}(t)$ (or an interpolation of its samples). Therefore, the value $x(0)$ estimated by the recognition model depends on the data, not on the output of the NODE. This has been corrected in the revised version.
> + The direct method with $t_c=0$ is indeed not very reasonable from a system theoretic point of view, as it assumes the system is instantaneously observable. However, many of the "augmentation strategies" used in NODEs boil down to it (Dupont et al, 2019; Massaroli et al, 2020b; Chalvidal et al, 2021; Norcliffe et al, 2021b).
> + When running a KKL observer in forward time, and if the analytical $\mathcal{T}^*$ is known, we have $\lim_{t \to \infty} \left| \mathcal{T}^*(z(t)) - x(t) \right| = 0$. Hence, if $t_c$ is indeed large enough and if $\mathcal{T}^*$ has been perfectly approximated, we do expect $x(0) \simeq \mathcal{T}^*(z(0))$ for the backward KKL recognition model.
> + For the exoskeleton data, the available prior model is quite poor: the predictions are worse with the prior model than with the initial, random guess on $f_\theta$. Also, when learning its residuals, we (slightly) penalize the difference between prior and current model. This explains why Fig. 7d is actually worse than 7b and 7c. However, we still wanted to show it to demonstrate that the proposed framework can be used to refine a given prior model with partial observations.
> + The proposed recognition models can indeed be applied to other machine learning tasks, such as image classification or general time series analysis. However, we are interested in state-space model identification, for which KKL recognition is in our opinion most suitable, since this is when its theoretical guarantees apply, and for which some degree of prior knowledge is available.
>
> We thank the reviewer for the other comments and have implemented them in the revised version. Changes are indicated in blue in the updated draft.

---

### Review · Reviewer_yCgK · 2022-11-17

**Summary Of Contributions:**

The paper proposes a model learning technique that can learn/estimate the initial hidden state. To estimate the initial state, the paper uses a KKL-based recognition model. In the experiments, the proposed method is applied to learn dynamics models on three different datasets, the earthquake model, the FitzHugh-Nagumo neuron model, the van der pol oscillator and an exoskeleton.

**Audience:**

Yes

**Claims And Evidence:**

Yes

**Requested Changes:**

* Improve the clarity of the proposed method and underlying theory.

* Make the structure of the dynamics model much clearer. Right now one can easily miss that the authors nearly assume to know the complete dynamics model. Framing some of the experiments as NODE is questionable as this overlaps with classical system identification methods.

* Improve the experiments to highlight the impact of the work. Right now the main results of the trivial toy problems are not very convincing. On the exoskeleton the importance of the results is questionable. It would be great to look deeper into how this model imrpoves the control of the exoskeleton. Reporting offline RMSE on fixed datasets has barely any implications for real-world control.

* All tables are missing confidence intervals.

**Strengths And Weaknesses:**

The strength of the paper is that it does not make outrageous claims and all statements are correct (deducting the normal overstatement). However, the paper has also many shortcomings.  The biggest problem is the clarity and hence the reproducibility of the paper. While the paper spends a lot of space on general approaches and related work, the actual contribution is very shallow. For the TLMR community, the presented knowledge of sections 1- 3 mostly exists. However, the TLMR community mainly consisting of ML researchers is not so fluent with observers and Kazantzis-Kravaris/Luenberger (KKL) observers, Therefore, it would be better to spend much more space on KKL observers such that the reader can understand the proposed method. The second problem is the experiments. Many of the experiments are very low dimensional and have been solved by hundreds of methods, which all beat some baselines (especially the van der pol oscillator). The exoskeleton is better from the complexity, but the reporting of single trajectories and the reported RMSE is not informative. It would be great to look at where this model is important and enables something new. The reported RMSE numbers are also not vastly better than the baselines.

---

> ### Author Response · Authors · 2022-11-24
> **Replies to review**
>
> We thank the reviewer for their comments. We hope that the changes we made (in blue in the revised draft) serve to improve the clarity of the proposed method and underlying theory. Below are answers to specific remarks.
>
> *Remark: For the TLMR community, the presented knowledge of sections 1- 3 mostly exists. However, the TLMR community mainly consisting of ML researchers is not so fluent with observers and Kazantzis-Kravaris/Luenberger (KKL) observers.*
>
> We thank the reviewer for their remarks on length and audience.
> As discussed in the response to Reviewer SjQP, we have reformulated the introduction and expanded Section 4.2 to make the concepts involved in KKL observers clearer for the TMLR audience.
> We have also slightly shortened sections 2 and 3 accordingly.
> However, we believe the related work should cover the two areas combined in our approach, namely nonlinear observer design and machine learning for dynamical systems, as stressed by Reviewer SjQP.
> Still, we are happy to consider concrete suggestions for further condensing.
>
> *Remark: Make the structure of the dynamics model much clearer. Right now one can easily miss that the authors nearly assume to know the complete dynamics model. Framing some of the experiments as NODE is questionable as this overlaps with classical system identification methods.*
>
> We have three sets of experiments: a comparison of recognition models, with three systems (earthquake model, FitzHugh-Nagumo, Van der Pol), a numerical study with varying degrees of prior knowledge (harmonic oscillator) and a real-world experimental dataset (exoskeleton data).
> For the recognition benchmark, we use a parametric model with unknown parameters that are optimized jointly. We had chosen this setting in order to have a ground truth to compare against, i.e. to ensure that the coordinate system for $x(t)$ is fixed in all experiments, and to focus on the recognition and not the model learning part.
> In light of the reviewer's remark, we have now run this benchmark again with full NODEs for each system instead of a parametric model. The revised manuscript contains these updated results, where the output estimation error is shown instead of the full state estimation error, due to each experiment finding a different set of coordinates. The previous results with a parametric model have been moved to the Appendix. Other than that, all experiments use full NODEs with varying degrees of prior knowledge, as described in the experimental section.
>
> *Remark: The second problem is the experiments. Many of the experiments are very low dimensional and have been solved by hundreds of methods, which all beat some baselines (especially the van der pol oscillator). The exoskeleton is better from the complexity, but the reporting of single trajectories and the reported RMSE is not informative.*
>
> These experiments serve two purposes. The benchmark systems and the harmonic oscillator are mainly illustrative, hence, we choose systems that are well known to the community. These systems have indeed often been considered, but rarely with partial and noisy observations, which is the main focus of this work.
> The exoskeleton experiments demonstrate that the proposed method can be used with real-world data. We stress that it is rare to find experimental robotics data in NODE papers, and that we compare the learned NODE against the linear model currently used for control at Wandercraft in both predicted test trajectories and with measurement injection through an observer.
> Finally, our focus in this paper lies on learning state-space models, which are common in control theory and allow for physical interpretation. This has now been made clearer in the introduction, in accordance with the remarks of Reviewer Gp37. Hence, we focus on such systems in state-space form with different degrees of prior knowledge in our experiments.
>
> *Remark: All tables are missing confidence intervals.*
>
> We have replaced the mean by the median and interquartile range in Table 2. For the exoskeleton data, we have run only one model per method due to limited computational resources.

---

> > ### Comment · Reviewer_yCgK · 2022-12-19
> > **Post Rebuttal Comment**
> >
> > Thanks for the revision, section 4.2 is much better. It would be great, if the authors could:
> >
> > * make the caption of figure 1 and 2 more concrete and include how the RSME is measured. Yes, it is written in the text but it would be great to not need to search for it.
> >
> > * include the sentence why the RSME on predicted trajectories is the right measure to compare these models. Yes, the reader should be smart enough to infer why but just make it obvious.
> >
> > While the paper is in an acceptable state now, the authors could do much more. You could include a very intuitive graphics that explains your process and where information flows forward and backwards in time. Furthermore, the 2d systems, the neuron and the van-der-pol oscilator, give you very nice opportunities for qualitative figures. You could plot the observed trajectory, the true trajectory and the predicted different initial conditions and the predicted trajectories. This would highlight much more how much better your proposed work is and give me further confidence in the work.

---

> > > ### Author Response · Authors · 2022-12-19
> > > **Reply to comment**
> > >
> > > We thank the reviewer for going over the revision. We have made the legends clearer and explained the RMSE more in detail (beginning of Sect. 5).
> > >
> > > We thank the reviewer for these suggestions. Graphs of true and predicted test trajectories for the recognition model benchmark are given in the supplementary material, in Fig. 9, 11 and 13. We have added the observed trajectory on this graph to give more intuition and show the effect of noise. However, each of these graphs is just one random test trajectory, and we believe a quantitative comparison with error plots is more suitable to compare different recognition models. Also, these graphs of the full state trajectory only make sense for the parametric model: for the full NODE model, each run finds a different set of coordinates, such that only the output trajectory is comparable.
> > > Finally, including these graphs in the main body of the paper would lead us to exceed the 12 page limit for regular submissions, therefore we keep them in the supplementary material.
> > > We believe Table 1 plays the role of an intuitive explanation of the different recognition methods, but will look into designing a more graphic approach.

---

> > > > ### Comment · Reviewer_yCgK · 2022-12-19
> > > > **Reply**
> > > >
> > > > The captions are good now.
> > > >
> > > > RE: Figures: It does not matter for acceptance anyways but your argument is very shortsighted. I am just stating my personal opinion that the figures and visualization can be improved.
> > > >
> > > > Yes, there is a page-limit of 12 pages but you can easily fit a lot more by just shifting focus between different aspects and rethinking how things are presented but you do not want to make the effort. Also the plots in the appendix are very uninspiring, you could be much more creative and rethink how this can be efficiently presented. You can be much more creative. The problem with the different coordinate systems can also be solved. Yes the naive version does not work but I believe you could find a solution for this.
> > > >
> > > > The authors have made the point now multiple times that they are not interested in improving the paper but want to solve the optimization problem of minimizing effort under the constraint of getting the paper accepted, which is a fair optimization objective. We have solved this optimization problem now and everything is good.

---

### Decision · Action_Editors · 2022-12-22

**Recommendation:** Accept with minor revision

**Comment:**

The paper proposes a new recognition model for latent Neural ODEs in the theme of proposing end-to-end learning physical models from experimental data. The proposed method utilizes the theory of  non-linear observer design (KKL observer) and builds a recognition model on top of it. With various empirical analysis the authors demonstrate the effectiveness of the proposed recognition network.

While some reviewers had concerns on capabilities and presentation, overall all of them stated claims in the paper are correct and supported by the evidence and the result would be relevant to some of the TMLR audience.

There were particular requests to improve visual presentation to guide the reader better.  The AE believes that the 12 page limit is not strict for TMLR (indeed longer paper is welcome and 12 page is just for guidance on the review timeline) and should be a soft suggestion if there's a tradeoff for clarity. Incorporating reviewer `yCgK`'s suggestion of including intuitive graphics for qualitative understanding of the method in the main body of the paper would be useful. While the paper is in a good state to be accepted, AE hopes to see some effort in improving clarity thus recommending "Accept with minor revision" for the camera-ready.


**Audience:**

As reviewer `yCgK` pointed out, most of the audience of TMLR would be ML based and would benefit from gentle introduction to observers and Kazantzis-Kravaris/Luenberger (KKL) observers to appreciate the contribution of the paper.

In general end-to-end learning of physical models from partially observable empirical data is an important topic that interests many of the TMLR audience.


**Claims And Evidence:**

Reviewers pointed out that claims are correct and empirical analysis of the paper indeed supports the utility of the proposed method. Here's quotes from the review and discussions.

- "paper certainly contributes to the development of a recognition model for neural ODEs"
- "the paper provides at least proof of the concept for the proposed method"
- "construction of the KKL observer-based recognition networks, is interesting."
- "experiments indeed show the possible utility of the proposed method."
- "experiments are very elaborate. "
- "KKL-based encoder almost always achieves better results or is on par. This is one of the strongest aspects of the work!"
- "Applying a KKL-based recognition model to latent ODEs or partially observed dynamical systems is a very natural idea."
- "the only neural ODE (NODE) encoder based on rigorous theory, which in turn comes with certain guarantees"

Some reviewers pointed out potential shortcomings:
- "biggest problem is the clarity and hence the reproducibility of the paper. "
- "actual contribution is very shallow. "
- "discussion about structured NODEs remains somewhat elementary"
- "experiments are very low dimensional and have been solved by hundreds of methods"
- "the visual presentation could be drastically improved such that the reader has more confidence and intuitive understanding of the results"
- "interrelation between the two topics of the paper (i.e., recognition models and structured NODEs) is unclear."

During the discussion period, the authors did a great job of addressing issues raised by the reviewers. The draft has undergone some non-trivial changes to address reviewers' concerns.

---

> ### Author Response · Authors · 2023-01-09
> **Reply to the AE**
>
> We thank the AE for their thorough and positive review. We followed the suggestions by the AE and included an intuitive graphical illustration of the method as Fig. 2, which should help with the qualitative understanding.
>
> We have also updated the plots for the FitzHugh-Nagumo and Van der Pol systems as suggested by reviewer yCgK, adding a phase portrait and a comparison between different recognition models (Fig. 12 and 14). We kept them in the supplementary material for brevity and because they only concern the parametric models, which have been replaced by full NODE models in the main body of the paper as originally suggested by reviewer yCgK.
>
> We have also changed the hyperref links to blue instead of a green box, as in paper https://openreview.net/pdf?id=cxp7n9q5c4, but we can revert the change if needed.
>
> We have uploaded a camera-ready version accordingly.